# Dose-dependent action of the RNA binding protein FOX-1 to relay X-chromosome number and determine *C. elegans* sex

**Behnom Farboud[1,2], Catherine S Novak[1,2†], Monique Nicoll[1,2‡], Alyssa Quiogue[1,2], Barbara J Meyer[1,2*]**

[1]Howard Hughes Medical Institute, University of California at Berkeley, Berkeley, United States; [2]Department of Molecular and Cell Biology, University of California at Berkeley, Berkeley, United States

**Abstract** We demonstrate how RNA binding protein FOX-1 functions as a dose-dependent X-signal element to communicate X-chromosome number and thereby determine nematode sex. FOX-1, an RNA recognition motif protein, triggers hermaphrodite development in XX embryos by causing non-productive alternative pre-mRNA splicing of *xol-1*, the master sex-determination switch gene that triggers male development in XO embryos. RNA binding experiments together with genome editing demonstrate that FOX-1 binds to multiple GCAUG and GCACG motifs in a *xol-1* intron, causing intron retention or partial exon deletion, thereby eliminating male-determining XOL-1 protein. Transforming all motifs to GCAUG or GCACG permits accurate alternative splicing, demonstrating efficacy of both motifs. Mutating subsets of both motifs partially alleviates non-productive splicing. Mutating all motifs blocks it, as does transforming them to low-affinity GCUUG motifs. Combining multiple high-affinity binding sites with the twofold change in FOX-1 concentration between XX and XO embryos achieves dose-sensitivity in splicing regulation to determine sex.

**\*For correspondence:**
bjmeyer@berkeley.edu

**Present address:** [†]Environmental Genomics and Systems Biology Division, Lawrence Berkeley National Laboratory, Berkeley, United States; [‡] Genentech, South San Francisco, United States

**Competing interests:** The authors declare that no competing interests exist.

## Introduction

Determining sex is one of the most fundamental developmental decisions that most organisms must make. Sex is often specified by a chromosome-counting mechanism that distinguishes one X chromosome from two: 2X embryos become females, while 1X embryos become males (*Bull, 1983*; *Charlesworth and Mank, 2010*). The nematode *Caenorhabditis elegans* determines sex with high fidelity by tallying X-chromosome number relative to ploidy, the sets of autosomes (X:A signal) (*Madl and Herman, 1979*; *Nigon, 1951*). The process is executed with remarkable precision: embryos with ratios of 1X:2A (0.5) or 2X:3A (0.67) develop into fertile males, while embryos with ratios of 3X:4A (0.75) or 2X:2A (1.0) develop into self-fertile hermaphrodites. Here we dissect molecular mechanisms by which the X:A signal specifies sex and thereby discover how small quantitative differences in intracellular signals can be translated into dramatically different developmental fates.

The X:A signal determines sex by controlling the activity of its direct target, the master sex-determination switch gene *xol-1* (XO lethal) (*Figure 1*; *Carmi et al., 1998*; *Farboud et al., 2013*; *Meyer, 2018*; *Miller et al., 1988*; *Nicoll et al., 1997*; *Powell et al., 2005*; *Rhind et al., 1995*). *xol-1* encodes a GHMP kinase that must be activated to set the male fate and repressed to set the hermaphrodite fate (*Luz et al., 2003*; *Rhind et al., 1995*). *xol-1* controls not only the choice of sexual fate but also X-chromosome gene expression through the process of X-chromosome dosage compensation (*Meyer, 2018*; *Miller et al., 1988*; *Rhind et al., 1995*). Males and hermaphrodites

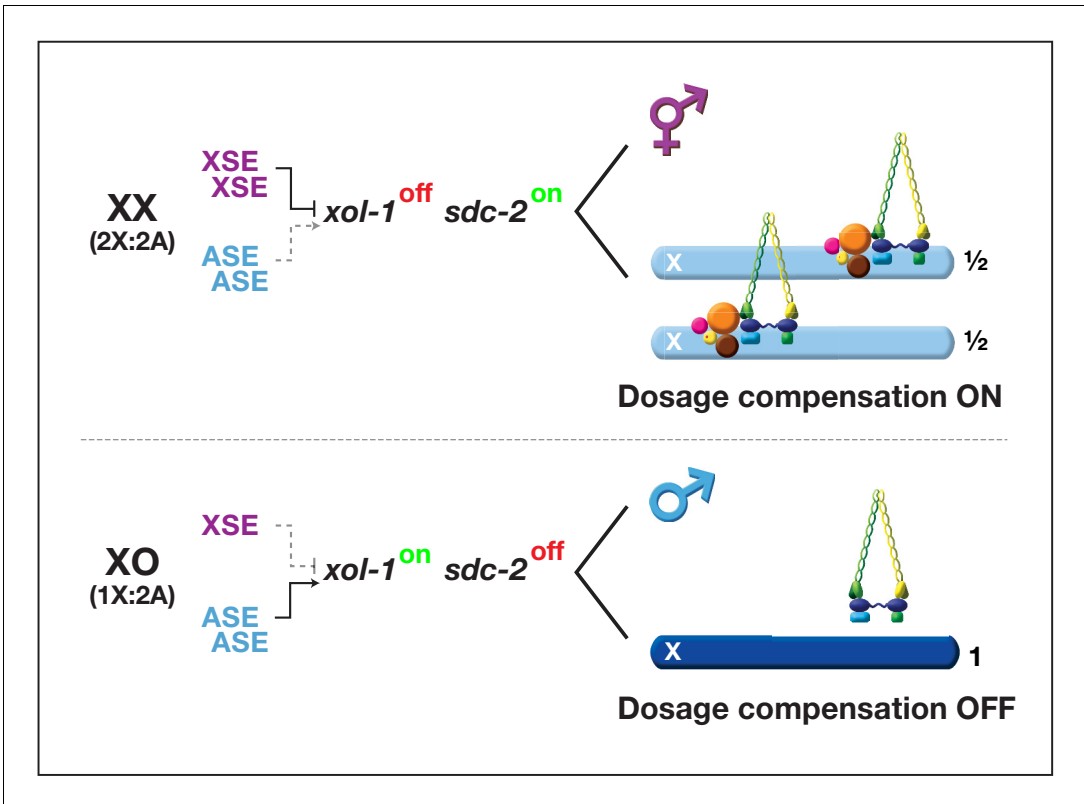

**Figure 1.** Overview of the X:A signal and regulatory hierarchy that control sex determination and X-chromosome dosage compensation. The X:A signal includes a set of genes on X called X-signal elements (XSEs) that repress their direct target gene *xol-1* (XO lethal) in a cumulative dose-dependent manner via transcriptional and post-transcriptional mechanisms and set of genes on autosomes called autosomal signal elements (ASEs) that stimulate *xol-1* transcription in a cumulative dose-dependent manner. *xol-1* is the master sex-determination switch gene that must be activated in XO animals to set the male fate and must be repressed in XX animals to permit the hermaphrodite fate. *xol-1* triggers male sexual development by repressing the feminizing switch gene *sdc-2* (sex determination and dosage compensation). *sdc-2* induces hermaphrodite sexual development and triggers binding of a dosage compensation complex (DCC) to hermaphrodite X chromosomes to repress gene expression by half. *xol-1* mutations enable the DCC to bind to the single male X chromosome and thereby kill all XO animals by causing reduced X-chromosome expression. The dying *xol-1* XO mutant animals are also feminized. Hence, mutations that disrupt elements of the X:A signal transform sexual fate and also alter X-chromosome gene expression.

generally require the same level of X-encoded gene products despite their difference in dose of X chromosomes. A dosage compensation complex (DCC) binds to both X chromosomes of diploid XX hermaphrodites to reduce transcription by half and thereby equalize X-linked gene expression with that from the single X of diploid XO males (*Figure 1*; *Chuang et al., 1994*; *Chuang et al., 1994*; *Csankovszki et al., 2009*; *Dawes et al., 1999*; *Lieb et al., 1998*; *Mets and Meyer, 2009*; *Mets and Meyer, 2009*; *Meyer, 2018*; *Pferdehirt et al., 2011*; *Tsai et al., 2008*; *Tsai et al., 2008*; *Wheeler et al., 2016*). The DCC resembles condensin, a chromosome remodeling complex required for the compaction and resolution of mitotic and meiotic chromosomes prior to their segregation during division (*Hirano, 2016*; *Meyer, 2018*; *Yatskevich et al., 2019*). Failure to activate dosage compensation kills all hermaphrodites. If *xol-1* is inappropriately activated in diploid XX animals, the DCC cannot bind to X chromosomes, and all hermaphrodites die from elevated X expression. Conversely, if *xol-1* is inappropriately repressed in diploid XO males, the DCC binds to the single X chromosome and kills all males by reducing X expression. Thus, the X:A signal determines the choice of sexual fate and sets the level of X-chromosome gene expression.

Our prior studies identified a set of genes on X chromosomes called X-signal elements (XSEs) that communicate X-chromosome dose by repressing *xol-1* in a dose-dependent manner

(*Figure 1*; *Akerib and Meyer, 1994*; *Carmi et al., 1998*; *Gladden et al., 2007*; *Gladden and Meyer, 2007*; *Hodgkin et al., 1994*; *Nicoll et al., 1997*). In addition, a set of genes on autosomes called autosomal signal elements (ASEs) communicates the ploidy by stimulating *xol-1* activity in a cumulative, dose-dependent manner to counter XSEs (*Figure 1*; *Farboud et al., 2013*; *Powell et al., 2005*). Two of the XSEs, the nuclear hormone receptor SEX-1 and the homeodomain protein CEH-39, as well as two of the ASEs, the T-box transcription factor SEA-1 and the zinc finger protein SEA-2, bind directly to multiple, non-overlapping sites in 5' transcriptional regulatory regions of *xol-1* (*Farboud et al., 2013*). XSEs and ASEs antagonize each other's opposing transcriptional activities to control *xol-1* transcript levels. The X:A signal is thus transmitted in part through multiple antagonistic molecular interactions carried out on a single promoter to regulate transcription (*Farboud et al., 2013*). Fidelity of X:A signaling is enhanced by a second tier of dose-dependent *xol-1* repression, via the XSE called FOX-1 (Feminizing locus On X), an RNA binding protein that includes an RNA recognition motif (RRM) (*Akerib and Meyer, 1994*; *Hodgkin et al., 1994*; *Nicoll et al., 1997*; *Skipper et al., 1999*).

The cumulative, dose-dependent action of XSEs was revealed by key genetic observations. For example, deleting one copy of *ceh-39* and *sex-1* from XX animals caused no lethality, but deleting one copy of *ceh-39*, *sex-1*, and *fox-1* killed more than 70% of XX animals, and deleting one copy of all genetically identified XSEs killed all XX animals (*Akerib and Meyer, 1994*; *Carmi and Meyer, 1999*; *Farboud et al., 2013*; *Gladden et al., 2007*). In reciprocal experiments, duplicating one copy of *fox-1* killed 25% of XO animals, while duplicating *fox-1* and *ceh-39* killed 50% of XO animals, and duplicating one copy of all genetically identified XSEs killed all XO animals (*Akerib and Meyer, 1994*; *Carmi and Meyer, 1999*; *Nicoll et al., 1997*).

Our current study analyzes the mechanism of FOX-1 action in regulating *xol-1. fox-1* was discovered originally through a mutation that suppressed the XO lethality caused by a large X duplication shown later to include multiple XSEs (*Akerib and Meyer, 1994*; *Hodgkin et al., 1994*; *Nicoll et al., 1997*). FOX-1 is the founding member of an ancient family of sequence-specific RNA binding proteins that are conserved from worms to humans (*Akerib and Meyer, 1994*; *Conboy, 2017*; *Hodgkin et al., 1994*; *Nicoll et al., 1997*). Recent experiments show that mammalian FOX family members recognize and bind the primary motifs GCAUG and GCACG but also bind secondary motifs with lower affinity (*Auweter et al., 2006*; *Begg et al., 2020*; *Jangi et al., 2014*; *Jin et al., 2003*; *Lambert et al., 2014*; *Lee et al., 2016*; *Modafferi and Black, 1999*; *Underwood et al., 2005*). FOX proteins regulate diverse aspects of RNA metabolism, including alternative pre-mRNA splicing, mRNA stability, translation, micro-RNA processing, and transcription (*Carreira-Rosario et al., 2016*; *Chen et al., 2016*; *Conboy, 2017*; *Jin et al., 2003*; *Kim et al., 2014b*; *Lee et al., 2016*; *Ray et al., 2013*; *Wei et al., 2016*). FOX proteins act as developmental regulators in different tissues of many species, controlling neuronal and brain development (*Begg et al., 2020*; *Gehman et al., 2012*; *Gehman et al., 2011*; *Kuroyanagi et al., 2013*; *Lee et al., 2016*; *Shibata et al., 2000*; *Underwood et al., 2005*) as well as muscle formation (*Gao et al., 2016*; *Kuroyanagi et al., 2007*; *Kuroyanagi et al., 2006*; *Singh et al., 2014*; *Wei et al., 2015*). *C. elegans* FOX-1 controls sex determination by repressing *xol-1* activity through a post-transcriptional mechanism that acts on any residual *xol-1* transcripts present in diploid XX animals after *xol-1* repression by the XSE transcription factors (*Carmi and Meyer, 1999*; *Nicoll et al., 1997*; *Skipper et al., 1999*). The level of this regulation, whether controlling pre-mRNA splicing, mRNA stability, nuclear transport, or translation of *xol-1* RNA, had not been determined.

We demonstrate that FOX-1 represses *xol-1* in XX embryos by regulating alternative *xol-1* pre-mRNA splicing to inhibit formation of the mature transcript that is both necessary and sufficient for *xol-1* activity in XO embryos. By binding to multiple sites in intron VI using both GCAUG and GCACG motifs, FOX-1 causes either intron VI retention or directs the use of an alternative 3' splice site, causing deletion of exon 7 coding sequences. Either alternative splicing event prevents production of essential male-specific XOL-1 proteins in XX embryos. Experiments performed in vivo demonstrate that intron VI is both necessary and sufficient for FOX-1-mediated pre-mRNA splicing regulation at the endogenous *xol-1* locus and at *lacZ* reporters. FOX-1 RNA binding experiments performed in vitro demonstrate that FOX-1 binds to multiple GCAUG and GCACG motifs in intron VI. Genome editing of endogenous motifs coupled with functional assays in vivo demonstrate that mutation of different GCAUG and GCACG combinations reduces FOX-1-mediated repression, but only mutation of all motifs or transformation of all to low affinity GCUUG motifs blocks non-

productive alternative splicing and mirrors the effect on X:A signaling of an engineered *fox-1* deletion. Splicing regulation is dose-dependent: mutating one copy of *fox-1* or all binding motifs in one copy of *xol-1* kills XX animals sensitized by reduced XSE activity. In contrast, conversion of all endogenous intron VI motifs to either GCAUG or GCACG permits normal splicing regulation, indicating that GCACG motifs are as effective as GCAUG motifs in promoting FOX-1 binding and splicing regulation. Hence, the number of high-affinity motifs is critical. Utilizing multiple high-affinity binding sites to elicit alternative splicing amplifies the X signal by permitting the concentration of FOX-1 made from two doses of *fox-1* in XX embryos to reach the threshold level necessary to inhibit XOL-1 production.

## Results

### An in vivo assay to determine regions of *xol-1* essential for repression by FOX-1

Prior studies showed that the RNA binding protein FOX-1 determines sex by repressing *xol-1* via a post-transcriptional mechanism, but the molecular basis of this regulation was not established (*Carmi and Meyer, 1999*; *Nicoll et al., 1997*; *Skipper et al., 1999*). We therefore devised an assay to identify regions of *xol-1* necessary for repression by FOX-1. Our prior experiments showed that overexpression of FOX-1 by itself is sufficient to repress endogenous *xol-1* activity, causing XO embryos to adopt the hermaphrodite sexual fate and die from reduced X-chromosome expression triggered by binding of the DCC to the single X (*Nicoll et al., 1997*). Hence, our strategy to identify FOX-1 regulatory sites was to assay deletion derivatives of a *xol-1* transgene controlled by the native *xol-1* promoter for responsiveness to FOX-1 repression in strains lacking the endogenous *xol-1* gene.

The wild-type *xol-1* transgene included all *xol-1* genomic sequences, and an insertion of *gfp* sequences in frame at the first ATG codon of *xol-1*. Expression of wild-type transgenes and deletion derivatives was monitored in *xol-1*(null) mutant strains by functional assays of XOL-1 activity. XX animals are very sensitive to the dose of *xol-1*, and extra-chromosomal arrays carrying wild-type *xol-1* transgenes could only be established using a 20–30-fold lower concentration of *xol-1* DNA than typical for routine markers (see Materials and methods). Wild-type transgenes in all seven independent arrays exhibited proper sex-specific regulation: XO animals were rescued from lethality caused by the endogenous *xol-1*(null) mutation, and XX animals were viable (*Figure 2A*). The proportion of XX versus XO animals in each line was in agreement with the expected ratio (2:1) from the male-producing mutation *him-5(e1490)* present in all lines. However, the *xol-1*(+) transgenes were expressed in XX animals at a somewhat higher level than the endogenous *xol-1* gene: the seven XX array lines could only be maintained if both endogenous copies of *fox-1* were wild type. The observation that *fox-1* mutations kill XX animals carrying wild-type *xol-1* transgenes shows the need for stringent *xol-1* repression in hermaphrodites by both transcriptional and post-transcriptional mechanisms.

For XO animals with wild-type *xol-1* transgenes, excess FOX-1 protein expressed from an integrated array [*yIs44(fox-1)*] carrying multiple copies of the *fox-1*(+) gene was lethal (*Figure 2A*; *Nicoll et al., 1997*). No XO animals were viable in the seven lines that carried wild-type *xol-1* transgenes and expressed high FOX-1 levels. Although GFP fluorescence was XO-specific when produced from wild-type transgenes, it proved to be too insensitive a monitor for changes in *xol-1* activity and was not used as part of our subsequent assays.

Deletion derivatives of *xol-1* transgenes lacking FOX-1 regulatory sequences are predicted to be insensitive to repression by excess FOX-1, allowing *xol-1*(null) XO animals to be rescued and fully viable. Deletion derivatives lacking FOX-1 regulatory regions are also expected to kill XX animals or cause visible dosage compensation defects (Dumpy and Egg-laying defective phenotypes) due to lack of repression by FOX-1. The XX lethality and other dosage compensation phenotypes should not be suppressed by excess FOX-1. Thus, the phenotypic consequences of wild-type and deletion-derivative *xol-1* transgenes act as sensitive monitors of regulation by FOX-1.

### Intron VI is essential for FOX-1 to repress *xol-1*

We first assayed the effects on XX and XO animals of *xol-1* transgenes with different combinations of intron deletions to define FOX-1 regulatory regions (*Figure 2A–E*). The deletion derivatives were

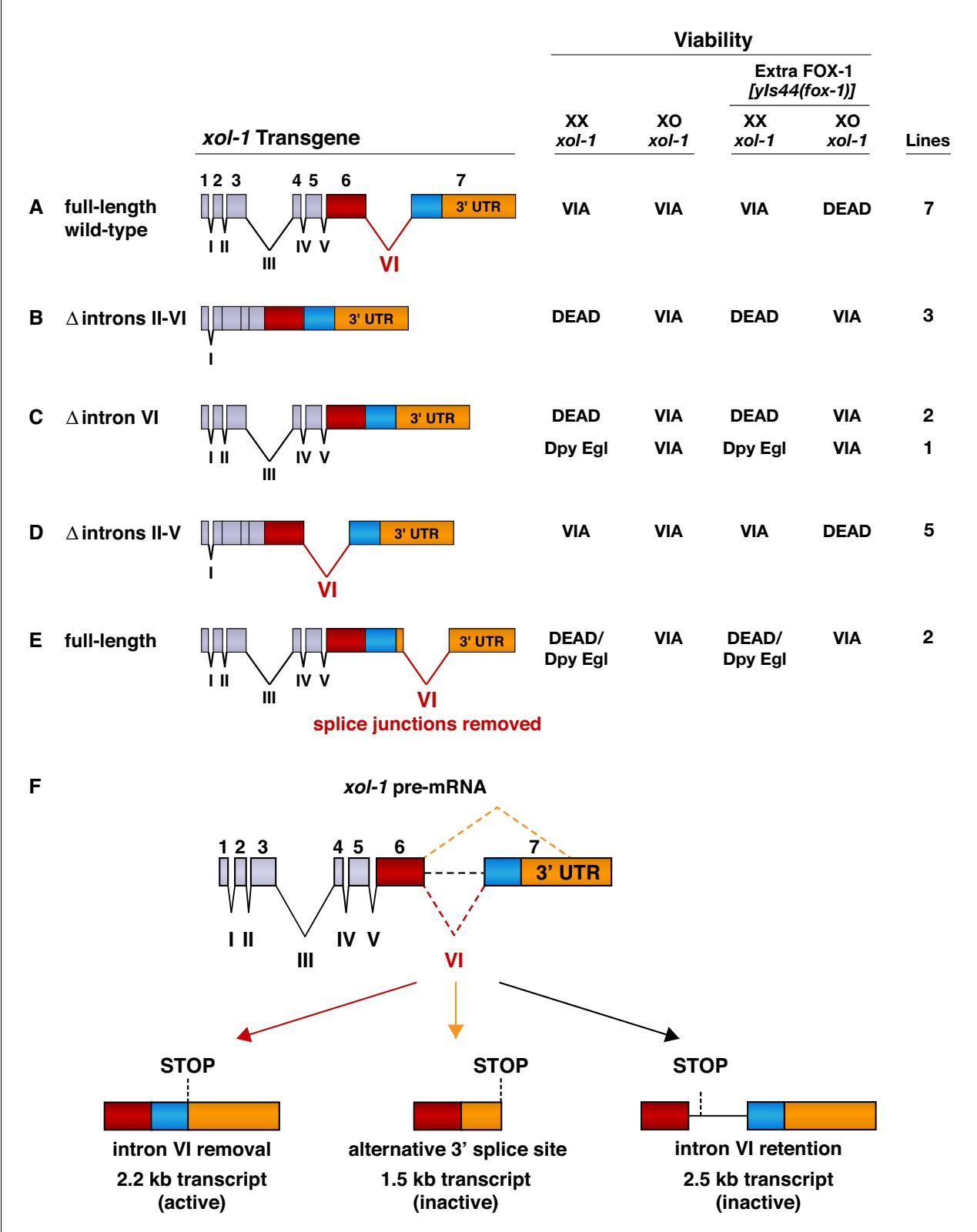

**Figure 2.** Intron VI is essential for repression of *xol-1* by FOX-1. (A–E) Assays of wild-type *xol-1* transgenes and deletion derivatives with different combinations of introns show that removal of intron VI prevents FOX-1 from repressing *xol-1*. Diagrams on the left show the intron–exon structure of *xol-1* sequences in the transgene derivatives. Exon 7 includes both coding sequences (blue) and the 3′ UTR (orange). The wild-type parent transgene rescues the XO-specific lethality caused by *xol-1* null mutations but permits XX animals to be viable. Intron deletion derivatives have genomic regions

*Figure 2 continued on next page*

Figure 2 continued

of *xol-1* replaced by corresponding *xol-1* cDNA to remove introns without altering protein coding sequence. Shown on the right is the viability of XX and XO *xol-1(y9)* deletion mutant animals carrying extra-chromosomal arrays of the different transgene derivatives. The arrays were made and assayed in *him-5(e1490); xol-1(y9)* mutants that produce 33% XO and 67% XX embryos and then crossed into *him-5; xol-1* strains producing high levels of FOX-1 from an integrated array *[yIs44 (fox-1)]* carrying multiple copies of *fox-1*. The far right shows the number of independent extra-chromosomal arrays assayed. The term 'dead' means that all animals of the genotype were inviable, and no extra-chromosomal arrays could be established in XX animals. The arrays could only be established and maintained through XO males. The term 'via' means the animals were viable and appeared wild type. The term 'Dpy Egl' refers to the phenotype of dosage-compensation-defective XX animals that escape lethality. XX animals are typically dumpy (Dpy) in body size and egg-laying defective (Egl). The term 'dead/Dpy Egl' means that most animals (greater than 90%) were dead, and rare escapers were Dpy Egl. High levels of FOX-1 kill all XO animals only if a spliceable form of intron VI is present. FOX-1 does not repress *xol-1* in XO animals if intron VI is absent or is relocated to the 3' UTR without splice junctions. Transgenes lacking intron VI kill XX animals, as do wild-type *xol-1* transgenes in strains with a *fox-1* null mutation, because all transgenes are expressed at a somewhat higher level than the endogenous *xol-1* gene in XX animals. (F) Structures of the three most abundant splice variants of *xol-1* transcripts are shown. Only the 2.2 kb variant, which lacks intron VI (red dashed line) and includes essential exon 7 coding sequences (blue) and 3' UTR, is necessary and sufficient for survival of XO animals. This isoform corresponds to Wormbase.org transcript C18A11.5b.1. Transcripts that retain intron VI (2.5 kb isoform corresponding to Wormbase.org transcript C18A11.5c.1) (black dashed line) or lack exon 7 coding sequences (1.5 kb isoform corresponding to Wormbase.org transcript C18A11.5a) (orange dashed line) due to use of an alternative 3' splice acceptor site cannot produce essential XOL-1 male-determining proteins.

The online version of this article includes the following source data for figure 2:

**Source data 1.** *xol-1* transcription is not repressed by high levels of FOX-1.

---

made by replacing genomic regions of *xol-1* with corresponding *xol-1* cDNA to remove introns without altering the protein coding sequence. Extra-chromosomal arrays carrying deletion derivatives of *xol-1* transgenes were created in *him-5; xol-1* strains capable of producing both XX and XO embryos and then crossed into *yIs44(fox-1); him-5; xol-1* strains that produce excess FOX-1. The DNA concentrations used for the distinct deletion-bearing transgenes were the same as for *xol-1*(+) transgenes. Even if a transgene deletion derivative causes XX-specific lethality, the array can be recovered and maintained through array-bearing XO animals.

The three array lines created from *xol-1* transgenes lacking introns II–VI (Δ introns II–VI) rescued *xol-1*(null) XO males, but killed all *xol-1*(null) XX hermaphrodites, even though the starting DNA concentration was the same as for *xol-1*(+) arrays (**Figure 2B**). Hundreds of array-bearing males were maintained for each independent array line through genetic crosses but no array-bearing hermaphrodite progeny survived, indicating that essential FOX-1 regulatory sequences had been deleted. Excess FOX-1 also failed to suppress the XX-specific lethality and failed to kill XO animals, demonstrating the necessity of intronic sequences for *xol-1* repression by FOX-1 (**Figure 2B**).

The essential role of introns in causing the death of XO animals with elevated FOX-1 levels could reflect the specific contribution of intron VI, which undergoes alternative splicing to yield at least three different *xol-1* transcript variants (**Figure 2F**; **Rhind et al., 1995**). Only the 2.2 kb variant, which lacks intron VI and includes all of exon 7, encodes a functional XOL-1 protein that has full XOL-1 activity essential for male development (**Figure 2F**). This 2.2 kb variant is both necessary and sufficient for full XOL-1 function in XO embryos (**Rhind et al., 1995**). It is the most abundant transcript of the three, accumulating to a level 10-fold higher in XO than XX embryos. A 1.5 kb variant is made from the same 5' donor in exon 6 as used for the 2.2 kb variant but a different 3' splice acceptor, one in the 3' UTR. This splicing event eliminates the coding region of exon 7, an essential exon (**Rhind et al., 1995**). The 1.5 kb variant does not encode *xol-1* XO activity. A 2.5 kb variant results from the failure to remove intron VI. In this variant, an in-frame UAA stop codon within intron VI terminates translation prematurely, precluding production of the male-determining protein (**Rhind et al., 1995**).

Indeed, removal of only the alternatively spliced intron VI from the *xol-1* transgene (Δ intron VI) in all three independent lines permitted all XO animals to be viable and to escape lethality caused by high levels of FOX-1 (**Figure 2C**). Two of the lines caused complete XX-specific lethality that was not suppressed by excess FOX-1; lines were maintained through genetic crosses with array-bearing XO males (**Figure 2C**). A third line caused milder dosage compensation defects in XX animals that were not suppressed by excess FOX-1 and caused the animals to be sterile, likely reflecting a lower copy number of the transgene and hence less expression. The line had to be maintained through array-bearing XO animals. Thus, intron VI is essential for FOX-1 repression of *xol-1*.

Further analysis confirmed that introns II–V are dispensable for FOX-1 regulation. All lines carrying transgenes lacking these introns (Δ introns II–V) behaved like lines of the wild-type *xol-1* transgene (*Figure 2D*). XX and XO *xol-1* animals in each line were viable in the relative proportion expected for the *him-5* mutation, and XO *xol-1* males were killed by excess FOX-1.

When intron VI was removed from its normal location in *xol-1* and relocated to the 3′ UTR without splice junctions, FOX-1 could not repress *xol-1* (*Figure 2E*). In two independent arrays of this transgene derivative, XO *xol-1* animals were viable, despite excess FOX-1, and greater than 99% of XX animals were inviable, even with excess FOX-1. The rare viable XX animals were Dpy and Egl. Thus, the presence of intron VI does not simply enable FOX-1 to repress *xol-1* by reducing mRNA stability, blocking nuclear transport of *xol-1* RNA or preventing translation in the cytoplasm. Furthermore, experiments quantifying levels of all *xol-1* RNA splice variants showed that high levels of FOX-1 did not diminish transcription from *xol-1* or cause transcript degradation (*Figure 2—source data 1*). Instead, FOX-1 likely regulates *xol-1* pre-mRNA splicing.

## FOX-1 regulates intron retention and alternative splicing of *xol-1* pre-mRNA

If FOX-1 represses *xol-1* by promoting alterative splicing, elevated levels of FOX-1 that cause male lethality should change the distribution of *xol-1* RNA splice variants. Specifically, the 2.2 kb splice variant that encodes male-determining activity should be reduced, while the inactive 2.5 kb variant with intron VI retention and/or the inactive 1.5 kb variant with partial exon 7 deletion should be increased. Using RNase protection assays and sequence analysis of cloned cDNAs, we show that elevated FOX-1 levels cause precisely these results (*Figure 3* and *Figure 3—figure supplement 1*).

For RNase protection assays, RNAs from two genetically distinct embryo populations were tested for the relative abundance of specific *xol-1* splice variants within each population: one RNA from *him-5* mixed-stage XX and XO embryos with wild-type FOX-1 levels and one RNA from *yIs44(fox-1); him-5* mixed-stage XX and XO embryos with high levels of FOX-1. These RNAs were assayed initially using an *act-1* control gene probe to assess the quality and quantity of RNAs (*Figure 3—figure supplement 1D*). The first set of *xol-1* RNase protection assays with these quantified RNAs used a *xol-1* antisense probe that spanned the 3′ splice junction of intron VI–exon 7 and distinguished the 2.2 kb and 2.5 kb transcripts (*Figure 3A*). The ratio of 2.2 kb and 2.5 kb splice variants was calculated from transcripts within each RNA population, and ratios from different populations were then compared to assess the change in transcript ratios caused by varying the FOX-1 level. XX and XO *him-5* mixed embryo populations with wild-type levels of FOX-1 showed a fourfold accumulation of active 2.2 kb transcript relative to the unspliced 2.5 kb transcript. In contrast, XX and XO populations that overexpressed FOX-1 [*yIs44(fox-1); him-5*] showed a drastic reduction in accumulation of the 2.2 kb transcript and a corresponding sixfold higher accumulation of the 2.5 kb transcript (*Figure 3A*). This experiment and repetitions (*Figure 3—figure supplement 1A*) established that FOX-1 promotes intron VI retention.

Transcript analysis using RT-PCR next showed that intron VI was the only intron retained in the presence of high FOX-1 levels. Using an oligonucleotide primer set that spans all six introns, RT-PCR was performed on cDNA made from the same RNAs as the protection assays. Only two PCR products were observed, one that corresponded to the 2.5 kb transcript and the other to the 2.2 kb transcript (see Materials and methods).

A second set of *xol-1* RNase protection assays utilized a different antisense probe that not only distinguished between the 2.2 kb and 2.5 kb transcripts, but also detected the 1.5 kb transcript and all alternatively spliced variants that contained the 3′ end of exon 6 (*Figure 3—figure supplement 1B*). The probe included the 3′ end of exon 6 and the 5′ end of exon 7. These protection experiments showed a decrease in accumulation of active 2.2 kb transcript relative to inactive 2.5 kb transcript in the presence of excess FOX-1 (2.4 to 1 with low FOX-1 and 1 to 5 with high FOX-1). They also showed a dramatic decrease in 2.2 kb transcripts compared to all spliced variants. With low FOX-1 levels, the 2.2 kb transcript was present at a ratio of 1:3, but with excess FOX-1 its accumulation decreased more than 10-fold, to a ratio of 1:38.

A third *xol-1* probe differentiated the inactive 1.5 kb transcript from the combination of 2.2 kb and 2.5 kb transcripts and from all alternatively spliced transcripts that included the 3′ end of exon 6 (*Figure 3—figure supplement 1C*). Excess FOX-1 caused the 1.5 kb transcript to increase in accumulation compared to the 2.2 kb and 2.5 kb transcripts, from a ratio of 1:14 with low FOX-1 to a

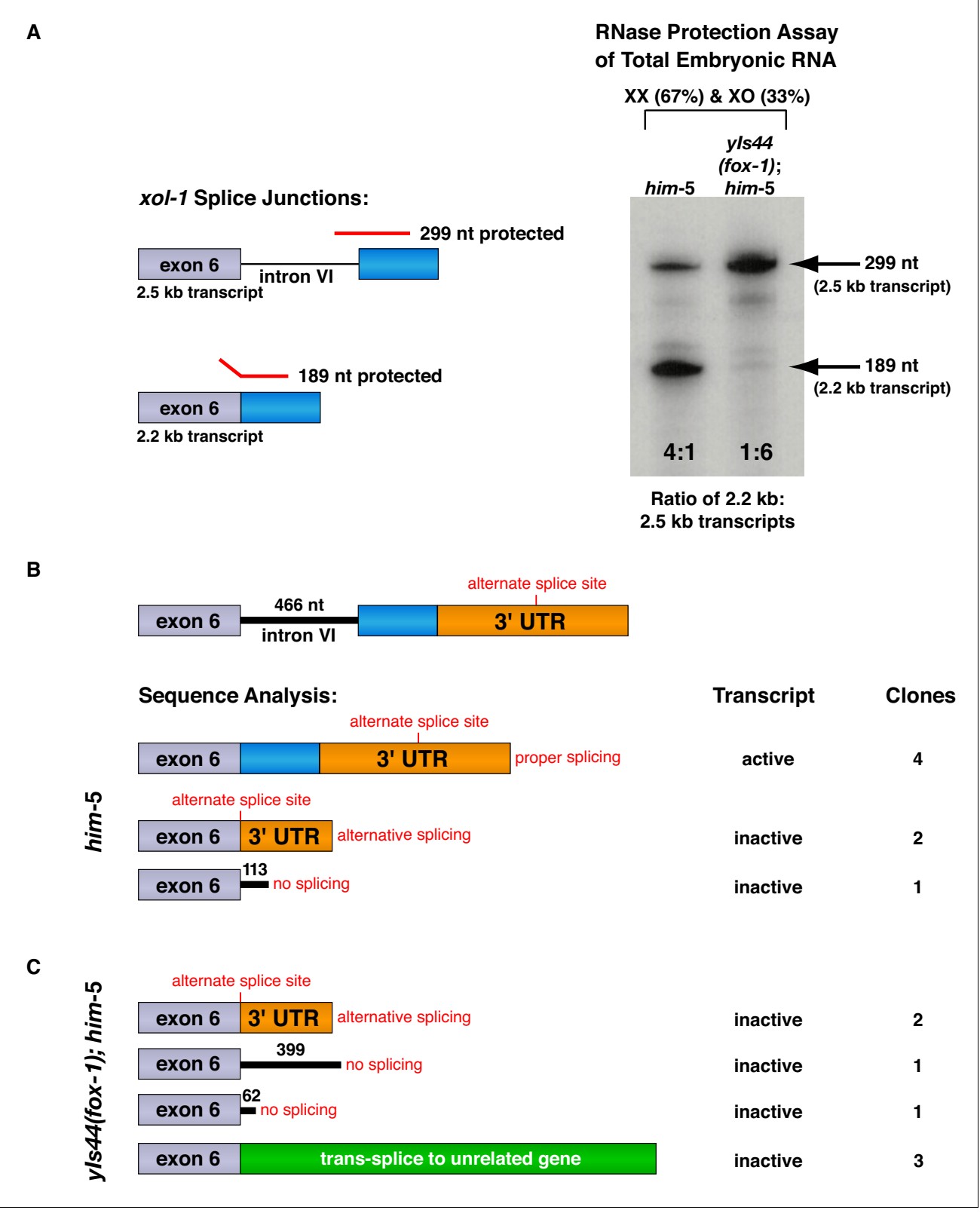

**Figure 3.** FOX-1 inhibits formation of the active 2.2 kb *xol-1* transcript by promoting intron retention and alternative 3′ splice acceptor selection. (A) RNase protection assays show that FOX-1 causes intron VI retention. Shown on the left are diagrams of relevant splice junctions for exon 6 (grey) and coding sequences of exon 7 (blue) that pertain to the inactive 2.5 kb and active 2.2 kb *xol-1* transcripts. Also shown are portions of the probe protected against RNase by each transcript: 299 nt for the 2.5 kb transcript and 189 nt for the 2.2 kb transcript. On the right is an image of the RNase protection

*Figure 3 continued on next page*

*Figure 3 continued*

assay of *xol-1* transcripts in *him-5* and in *yIs44(fox-1); him-5* strains quantified by a phosphorimager. The probe is labeled with [32]P-UTP: 43 U residues in the intron VI portion and 44 U residues in the exon 7 portion. Prior to quantifying the ratio of 2.2 kb to 2.5 kb transcripts, the 299 nt signal was divided in half to compensate for its higher number of U residues. The assay demonstrates that FOX-1 inhibits production of the male-determining 2.2 kb transcript by preventing the removal of intron VI. Levels of the *act-1* control transcript were also assayed in these two RNA samples (***Figure 3—figure supplement 1D***). Quantification of a separate RNase protection experiment using this pMN86 probe is shown in ***Figure 3—figure supplement 1B***. (B, C) Sequence analysis of cDNA clones from *xol-1* transcripts shows that FOX-1 causes intron VI retention and also alternative 3′ splice acceptor selection in *xol-1*. FOX-1 also causes *trans*-splicing of *xol-1* pre-mRNA to pre-mRNA of unrelated genes. Below the diagram of *xol-1*'s relevant intron–exon structure (exon 6 in grey and exon 7 coding sequences in blue) are diagrams representing the splicing pattern revealed by DNA sequence analysis of *xol-1* cDNAs from *him-5* (B) and from *yIs44(fox-1); him-5* strains (C). Also shown are the predicted *xol-1* activity states of transcripts with the different splicing patterns and the number of cDNA clones with each pattern. In instances of intron VI retention, the number of intron VI nucleotides in each clone is shown with blue numbers. During *trans*-splicing (C), the proper 5′ donor at the exon 6–intron VI junction was used together with a naturally occurring 3′ acceptor at an intron–exon junction of an unrelated gene identified in the text. Resulting *trans*-splicing events had the junction expected from proper use of the 5′ and 3′ sites.

The online version of this article includes the following figure supplement(s) for figure 3:

**Figure supplement 1.** RNase protection experiments show that FOX-1 promotes intron VI retention and partial exon 7 deletion to inhibit formation of the functional 2.2 kb *xol-1* transcript.

ratio of 1:4 with high FOX-1. The 1.5 kb transcript also increased in accumulation compared to all splice variants present in excess FOX-1, from a 1:35 ratio to a 1:21 ratio.

This series of protection experiments demonstrated that FOX-1 represses *xol-1* by controlling its pre-mRNA splicing, promoting both intron retention and also deletion of exon 7 coding sequences via alternative 3′ acceptor site choice. They also reveal that excess FOX-1 causes the production of more splice variants than previously mapped. Abundance of the universal protected fragment from the 3′ end of exon 6 (79 nt) relative to the protected fragments from known splice variants (191 nt and 144 nt) (***Figure 3—figure supplement 1A***) indicates the occurrence of unidentified splice variants caused by excess FOX-1. As a consequence, we took an alternative approach to identify splice variants by synthesizing cDNA from total embryonic RNA made from both the *him-5* and the *yIs44; him-5* strains and then selectively cloning and sequencing cDNAs that contained the 3′ end of exon 6 (***Figure 3B***).

Sequences of cDNA clones from *xol-1* transcripts in low FOX-1 conditions revealed only the expected: properly spliced active 2.2 kb transcripts, inactive 2.5 kb transcripts that retained intron VI, and inactive alternatively spliced 1.5 kb transcripts that deleted part of exon 7 through use of a 3′ acceptor site in the 3′ UTR (***Figure 3B***). Clones of cDNAs from high FOX-1 conditions revealed both expected and unexpected transcripts (***Figure 3C***). As expected, no active 2.2 kb transcripts were found, but splice variants that retained intron VI or deleted exon 7 were found. Unexpectedly, several cDNAs corresponded to *xol-1* transcripts that had been *trans*-spliced to unrelated genes. This splicing involved the 5′ donor at the exon 6–intron VI splice junction, permitting inclusion of exon 6, but then utilized an naturally occurring 3′ acceptor site at an exon from an unrelated gene on chromosome II, either *flcn-1*, *polyg-1*, or *K02E7.12*, thus achieving accurate *trans*-splicing. Thus, the process by which FOX-1 enhances intron retention during RNA processing also promotes the use of alternative 3′ acceptor sites, causing deletion of exon coding sequences and enabling *trans*-splicing.

Promiscuous *fox-1*-mediated *trans*-splicing that fuses partially spliced *xol-1* transcripts onto transcripts from unrelated genes may result from the normal 3′ acceptor site in *xol-1* exon 7 being unavailable to receive the primed 5′ splice donor in exon 6, perhaps because the 3′ site is blocked or not yet synthesized. The primed 5′ site then fuses with a nearby available 3′ acceptor site, whether or not the site is in *xol-1* or an unrelated gene. Analysis of recovered promiscuous splicing events revealed *xol-1* transcripts spliced not only to transcripts from chromosome II genes, but also, and even more frequently, to transcripts from adjacent genes encoded on transgenic arrays that include *xol-1*. Although *trans*-splicing between a common 22 nucleotide SL-1 or SL-2 leader sequence and the 5′ end of nascent transcripts has been well documented in *C. elegans* (***Blumenthal, 2012***), trans-splicing of exons from two different protein-coding genes has not been reported previously. In contrast, developmentally programmed trans-splicing has been observed in Drosophila

for neighboring genes (*Büchner et al., 2000*; *Dorn et al., 2001*; *Gabler et al., 2005*; *Horiuchi et al., 2003*; *Labrador et al., 2001*).

## Intron VI is sufficient for FOX-1-mediated repression

To understand the mechanism by which FOX-1 regulates intron splicing, we asked whether intron VI is sufficient to confer FOX-1 repression upon a heterologous transcript (*Figure 4A,B*). We placed *xol-1*'s 466 bp intron VI into the fifth exon of a *lacZ* reporter gene driven by the *xol-1* promoter and tested whether excess FOX-1 from *yIs44(fox-1)* could prevent its expression (*Figure 4B*). The reporter gene also included the 3′ UTR from the *unc-54* myosin gene to assess whether intron VI by itself, without the 3′ UTR from *xol-1*, can confer FOX-1-mediated repression. Four independently derived lines with extra-chromosomal arrays carrying the *Pxol-1::lacZ::intron VI::unc-54 3′ UTR* reporter exhibited proper sex-specific regulation, expressing β-galactosidase at high levels in XO embryos (*Figure 4B*, see XX and XO images) and low levels in XX embryos (*Figure 4B*, see XX image), as did five independently derived lines with control *Pxol-1::lacZ::xol-1 3′ UTR* reporter arrays that lacked intron VI and had a 3′ UTR from *xol-1* (*Figure 4A*, see XX vs. XX and XO images). Excess FOX-1 caused a marked decrease in the frequency and intensity of embryonic β-galactosidase expression in all four intron-VI-containing lines (*Figure 4B*, see *yIs44* image) but had no effect on the five lines carrying the control *lacZ* reporter without intron VI (*Figure 4A*, see *yIs44* image). At least 1000 embryos were scored for each genotype derived from each of the nine independent arrays. Therefore, intron VI by itself is sufficient to confer FOX-1 repression. These results also show that *xol-1*'s 3′ UTR is neither necessary nor sufficient for FOX-1 repression.

To determine whether repression of the *lacZ* reporter occurs by the same mechanism as repression of *xol-1*, we examined the splicing pattern of the reporter transcripts. cDNA was synthesized from total embryonic RNA made from intron VI-containing *lacZ* reporters expressed in both *him-5* and *yIs44(fox-1); him-5* strains. *lacZ*-specific PCR primer sets were used to clone *lacZ* transcripts from the two strains and determine the splicing pattern. Clones from XX animals with low FOX-1 levels revealed the same classes of RNA processing events in *lacZ* transcripts as found from RNA processing events of endogenous *xol-1* RNA, consistent with intron VI conferring FOX-1 repression (*Figure 4C*). They included a class with proper intron VI removal, one with intron VI retention, one with proper intron VI removal plus an additional splice in *lacZ* sequences corresponding to the DNA region between AgeI and PvuI restriction sites, and one with the correct 5′ donor site usage at the *lacZ*–intron VI junction but an alternative 3′ splice acceptor in an exon of an unrelated gene, *unc-76* (chromosome V), *F58D5.5* (chromosome I), or *H05L03* (chromosome X).

Clones from XX animals with high FOX-1 levels revealed five classes of transcripts consistent with transcripts from endogenous *xol-1* in the presence of high FOX-1, confirming that intron VI is sufficient for FOX-1 regulation (*Figure 4C*). The classes included one that had proper intron VI removal, one that retained intron VI, one that used the correct 5′ donor at the *lacZ*–intron VI junction but an alternative 3′ splice acceptor in *lacZ*, one that had aberrant 5′ and 3′ junctions in *lacZ*, and one that used the correct 5′ donor but a naturally occurring 3′ splice acceptor at an exon of an unrelated gene, most commonly *unc-76*, but also to *zen-4* (chromosome IV) and *T28F4.4* (chromosome I), an ortholog of human ARMC5. The high frequency of *trans*-splicing, particularly to *unc-76*, likely results from both *lacZ* and *unc-76* being transcribed from the same extra-chromosomal array. We conclude that repression by FOX-1 can be conferred in vivo onto a heterologous gene simply by the insertion of intron VI. Moreover, repression occurs by promoting either intron retention or use of alternative 3′ acceptor sites, resulting in deletion of exon coding sequences and enabling *trans*-splicing.

## FOX-1 binds directly to multiple sites in intron VI

Because intron VI is both necessary and sufficient for *xol-1* repression by FOX-1, we developed an in vitro assay to determine whether FOX-1 regulates RNA splicing by binding directly to intron VI. Using purified FOX-1 protein and equimolar amounts of $^{32}$P-labeled RNA probe for full-length intron VI and for intron III, we performed initial cross-linking binding assays to find conditions that could permit highly specific FOX-1 binding. In experiments with increasing concentrations of FOX-1, we found that FOX-1 bound robustly to the intron VI probe, but not to the intron III negative control probe (*Figure 5—figure supplement 1A*). Also, intron III RNA served as a better nonspecific competitor than tRNA to inhibit nonspecific binding (*Figure 5—figure supplement 1A*). In competition

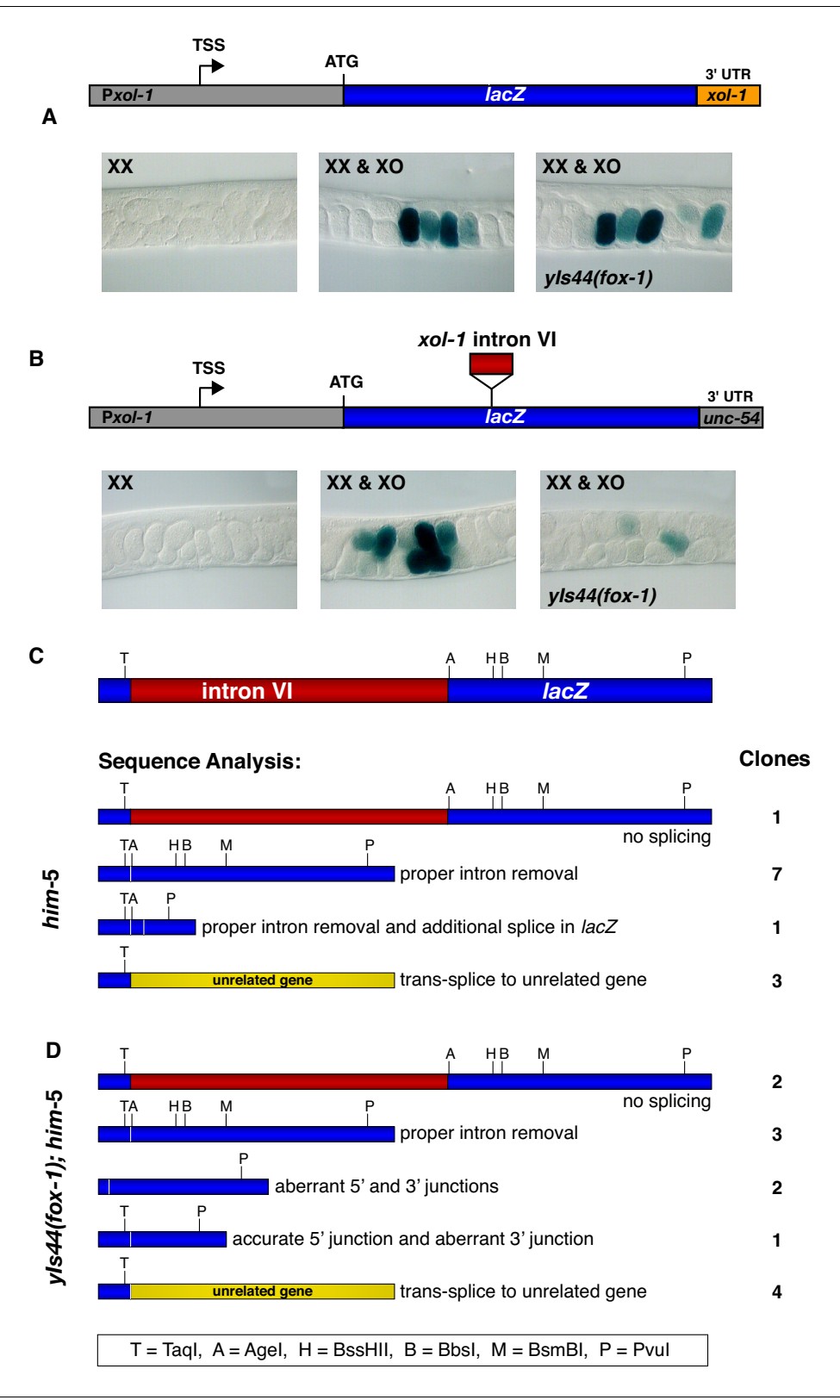

**Figure 4.** Intron VI of *xol-1* is sufficient to confer FOX-1 repression. (**A**) The promoter and 3′ UTR of *xol-1* are not sufficient for FOX-1 to repress *xol-1*. Below the diagram of the *Pxol-1::lacZ::xol-1 3′ UTR* reporter transgene (pMN27) are sections of adult gonads from different genotypes stained with 5-bromo-4-chloro-3-indolyl-D-galactopyranoside. Genotypes of embryos in the gonads include: (left) XX, *unc-76; yEx231* [pMN27 and *unc-76* (+)]; (middle) XX and XO, *him-5 unc-76; yEx231* [pMN27 and *unc-76* (+)]; (right) XX and XO, *yIs44(fox-1); him-5 unc-76; yEx231* [pMN27 and *unc-76* (+)]. The *lacZ*
*Figure 4 continued on next page*

*Figure 4 continued*

reporter is sex-specifically regulated: high levels of β-galactosidase in XO embryos but low levels in XX embryos. High levels of FOX-1 do not diminish β-galactosidase activity in the absence of intron VI, indicating that the *xol-1* promoter and *xol-1* 3' UTR cannot confer FOX-1 repression. Five independent extra-chromosomal array strains of each genotype carrying pMN27 showed the results represented. At least 1000 embryos were examined for each genotype derived from each of the five independent arrays. (B) Intron VI of *xol-1* is sufficient for FOX-1 to repress a *lacZ* reporter gene. Shown is a diagram of the *Pxol-1::lacZ::intronVI::unc-54 3' UTR* reporter transgene (pMN110) in which the 3' UTR is from the body-wall myosin gene *unc-54*. Genotypes of adult gonads stained for β-galactosidase activity are the same as listed in (A), except the array is *yEx280* [(pMN110) and *unc-76* (+)]. This intron VI-containing *lacZ* reporter is also sex-specifically regulated: active in XO embryos and repressed in XX embryos. High levels of FOX-1 (from *yIs44*) greatly diminish the level of β-galactosidase activity in XO embryos, indicating that intron VI alone is sufficient for FOX-1 repression. Four independent extra-chromosomal array strains of each genotype carrying pMN110 showed the results represented. At least 1000 embryos were examined for each genotype derived from each of the four independent arrays. (C, D) Sequence analysis of cDNA clones from *lacZ* transcripts shows that excess FOX-1 increases intron VI retention and also causes alternative pre-mRNA splicing using 3' splice acceptor sites in *lacZ* via *cis*-splicing and also 3' splice acceptor sites in unrelated genes via *trans*-splicing. Below the diagram of *lacZ*'s relevant intron–exon structure and restriction sites is the sequence analysis of *lacZ* cDNAs from *him-5* (C) and from *yIs44(fox-1); him-5* (D) strains. Shown are the splicing patterns revealed by DNA sequence analysis and also the number of *lacZ* clones with each indicated structure. During *trans*-splicing, the proper 5' donor at the *lacZ* exon–intron VI junction was used in combination with a naturally occurring 3' acceptor at an intron–exon junction of an unrelated gene (see text).

experiments for which $^{32}$P-labeled intron VI was challenged with increasing concentrations of either cold intron III RNA or cold intron VI RNA, intron III did not compete for FOX-1 binding. In contrast, cold intron VI severely reduced FOX-1 binding to intron VI probe (***Figure 5—figure supplement 1B***). These results show that FOX-1 binds directly and specifically to intron VI. All subsequent binding experiments were performed in the presence of cold intron III RNA to inhibit nonspecific binding.

Direct binding assays with $^{32}$P-labeled RNA probes to five subregions of intron VI demonstrated specific binding to three, fragments B, C, and E (***Figure 5—figure supplement 1C***). Competition experiments using intron VI RNA as probe and cold RNA fragments as competitors confirmed that FOX-1 binds to multiple sites in intron VI (***Figure 5—figure supplement 1D***). Sequence analysis of the FOX-1 binding fragments revealed common RNA sequences within the three fragments, as delineated by B-37, C-35, and E-35 in ***Figure 5A***.

To determine whether these common sequences are responsible for FOX-1 binding, we first used three RNA oligonucleotides of different sizes to fragment B in direct competition experiments against $^{32}$P-labeled fragment B probe and $^{32}$P-labeled intron VI probe (***Figure 5B,C***). FOX-1 binding to the B probe was eliminated not only by cold B RNA, but also by the cold 45 nt and 37 nt RNA oligonucleotides that covered the entire common fragment B sequence (***Figure 5B***). Thus, the common sequence is sufficient for FOX-1 binding. In addition, FOX-1 binding to $^{32}$P-intron VI probe was eliminated by both cold intron VI RNA and cold 45 nt and 37 nt fragment B RNA oligonucleotides, indicating the common sequence supports high-affinity binding that might account for all FOX-1 binding to intron VI (***Figure 5C,D***). Consistent with that interpretation, FOX-1 binding to $^{32}$P-intron VI was eliminated by a cold 35 nt RNA oligonucleotide to fragment C sequence and a cold 35 nt RNA oligonucleotide to fragment E sequence (***Figure 5D***).

In contrast, a 15 nt RNA oligonucleotide (CAUUUGAUCGUUAUG) from the middle of sequences common to all three FOX-1 binding fragments was incapable of competing for FOX-1 binding to either a fragment B probe or an intron VI probe, indicating that FOX-1 utilizes one or both of the small motifs, GCAUG and GCACG (***Figure 5B,C***). Both GCAUG and GCACG are in fragments B and C, but only GCACG is in fragment E.

A 25 nt RNA oligonucleotide that includes GCACG and the center of the common sequence in fragment E was sufficient to eliminate binding to an intron VI probe, indicating that GCACG promotes strong FOX-1 binding, and GCAUG is not essential for FOX-1 binding (***Figure 5D***). This result does not exclude the interesting possibility that GCAUG might substitute for GCACG or enhance binding to RNA that also includes GCACG.

In a final series of competition experiments, we asked whether FOX-1 binding to intron VI utilizes all three separate regions of common sequence. We compared a cold intron VI RNA competitor that lacked the common sequences in fragments B and C (Δ37 Δ35) with a cold intron VI RNA competitor that lacked the common sequences in fragments B, C, and E (Δ37 Δ35 Δ26) (***Figure 5D***). While the Δ37 Δ35 intron was a very poor competitor for the intact intron VI probe, the Δ37 Δ35 Δ26 intron was even worse; it lacked the ability to compete (***Figure 5D***). Thus, the three regions of common

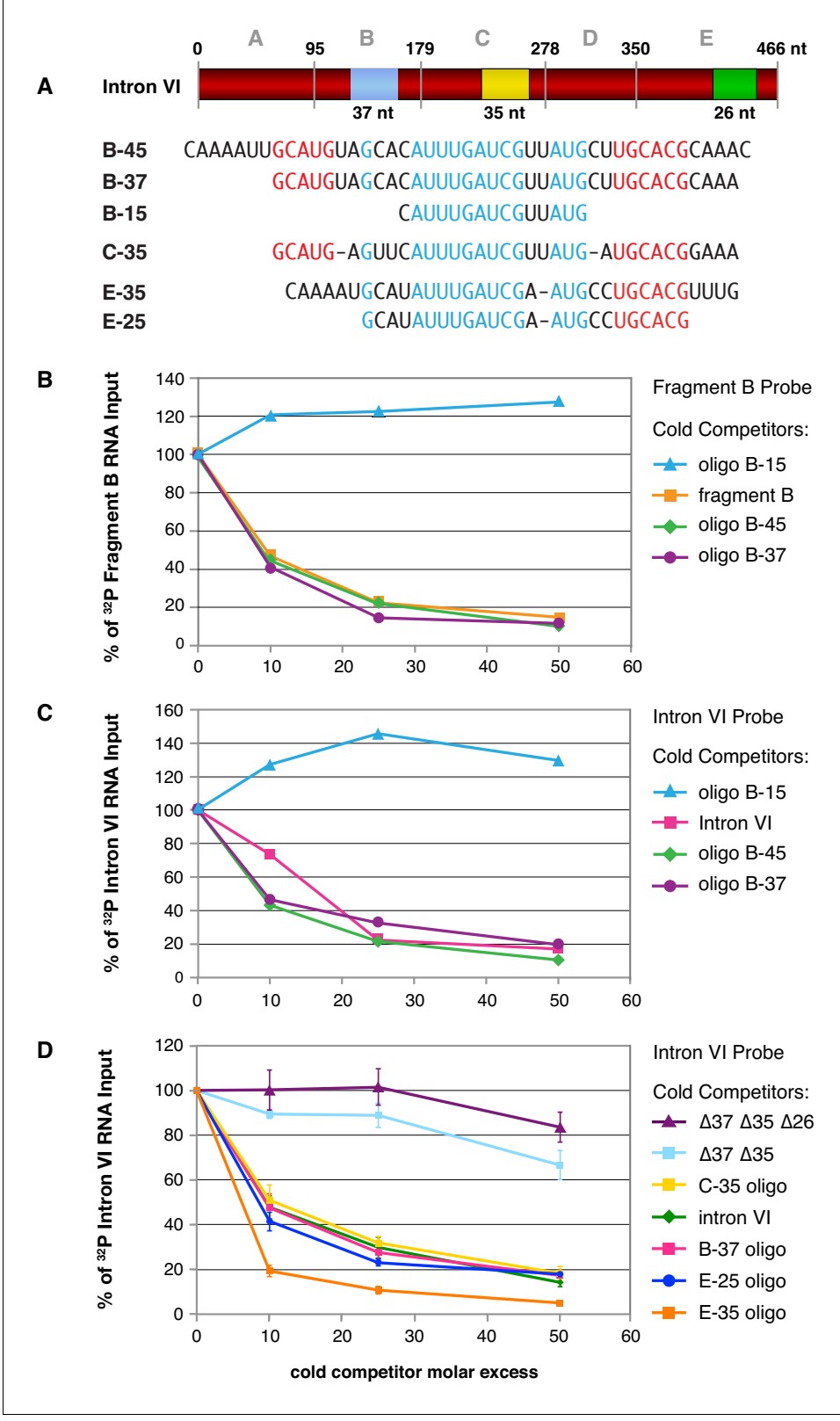

**Figure 5.** Purified FOX-1 protein binds in vitro to multiple sites in intron VI using motifs GCAUG and GCACG. (**A**) The diagram of intron VI shows the intron VI fragments (A–E) and smaller regions (RNA oligonucleotides B-45 to E-25) tested for direct FOX-1 binding in vitro. Only RNAs from fragments B, C, and E bind to purified FOX-1 (*Figure 5—figure supplement 1C*). Motifs GCAUG and GCACG (red) and sequences common to all three
*Figure 5 continued on next page*

*Figure 5 continued*

fragments (light blue) are shown below the diagram. Blue, yellow, and green rectangles indicate the locations of sequences B-37, C-35, and E-25, respectively, within FOX-1 binding fragments of intron VI. The RNA oligonucleotides listed were used in competition experiments with $^{32}$P-labeled fragment B (panel **B**) and $^{32}$P-labeled intron VI (panels **C** and **D**). (**B**) Small RNA oligonucleotides corresponding to sequences within fragment B compete for FOX-1 binding in vitro. Graphs show cross-linking competition experiments in which binding of FOX-1 (32 ng) to $^{32}$P-labeled fragment B RNA was challenged with an increasing molar excess of either cold fragment B RNA or small RNA oligonucleotides to sequences in fragment B that are also found in the other FOX-1 binding regions in fragments C and E. Binding is expressed as the percent of $^{32}$P fragment B bound by FOX-1 without any competitor RNA. (**C**) RNA oligonucleotides compete for FOX-1 binding to intron VI. The cross-linking competition experiments are similar to those in panel (**A**), except the probe is $^{32}$P-labeled full-length intron VI RNA. Binding is expressed as the percent of $^{32}$P intron VI bound by FOX-1 without any competitor RNA. The finding that the B-15 oligonucleotide fails to compete with either fragment B probe or intron VI probe, while the B-45 and B-37 oligonucleotides compete well, indicates that GCAUG, GCACG, or both are utilized for FOX-1 binding. (**D**) FOX-1 binds to multiple sites within intron VI using both GCAUG and GCACG. Graphs show results of cross-linking competition experiments in which binding of FOX-1 (32 ng) to $^{32}$P-labeled intron VI RNA was challenged with an increasing molar excess of several cold RNAs, as indicated. Binding is expressed as the percent of $^{32}$P intron VI RNA bound by FOX-1 without any competitor RNA. Cold intron VI RNA carrying deletions of the common sequences in B (Δ37) and C (Δ35) competed very poorly with intron VI probe for FOX-1 binding, and cold intron VI with deletions in all three common regions [B (Δ37), C (Δ35), and E (Δ26)] competed even less efficiently, demonstrating the critical role of these sequences in FOX-1 binding. In contrast, RNA oligonucleotides (C-35, B-37, E-35, and E-25) of sequences in fragments B, C, and E competed very effectively with intron VI for binding to FOX-1, further supporting the conclusion that FOX-1 binds to multiple sites in intron VI. The 25 nt RNA oligonucleotide in fragment E contains only the motif GCACG, but not GCAUG, indicating that GCACG promotes robust FOX-1 binding. The deletion in E (Δ26) is one nucleotide longer than the E-25 oligonucleotide, including deletion of a 3′ U. Error bars, SEM.

The online version of this article includes the following figure supplement(s) for figure 5:

**Figure supplement 1.** Cross-linking experiments show that FOX-1 binds directly to multiple sites in intron VI.

---

sequence contribute to FOX-1 binding in vitro and suggest they might all contribute to FOX-1 repression in vivo. The competition experiments reinforce the model that direct FOX-1 binding to intron VI facilitates intron VI retention and also causes deletion of exon 7 coding sequences by promoting use of an alternative 3′ splice acceptor site.

## Disruption of endogenous FOX-1 binding sites in intron VI abrogates splicing-mediated repression of *xol-1* in vivo

Identification of FOX-1 binding sites in vitro led us to analyze the function of these sites in vivo for regulating *xol-1* splicing and to determine the impact of *xol-1* splicing regulation on X-signal activity during normal nematode development (*Figure 6*). Thus far, our experiments identified intron VI as the target of *xol-1* splicing regulation in the context of elevated *xol-1* expression caused by multiple copies of *xol-1*. Because elevated *xol-1* expression is lethal to XX animals, these results cannot be extrapolated to disclose the full contribution of splicing regulation to X-signal activity during normal embryogenesis. The impact of splicing regulation in vivo can be determined by editing the endogenous *xol-1* gene to eliminate sites used in vivo for regulation by FOX-1.

The approach of removing *cis*-acting sites in *xol-1* by CRISPR/Cas9-mediated genome editing confers three additional advantages over mutating the *fox-1* gene itself for analyzing pre-mRNA splicing regulation. It cleanly separates the role of FOX-1 in sex determination from its roles in other developmental processes, revealing a more precise understanding for the contribution of alternative splicing regulation to the X signal. Furthermore, eliminating intron VI blocks all RNA binding proteins and accessory factors from participating in *xol-1* splicing, potentially revealing a larger role for splicing regulation in communicating the X signal than simply achieved by FOX-1 alone. Lastly, disrupting individual FOX-1 binding sites allows us to determine the specific sites used for splicing regulation and the number of sites needed to convey the effect of a twofold difference in FOX-1 dose between XO and XX embryos to specify sex. Repression through multiple sites has the potential to amplify the small change in FOX-1 concentration between sexes by minimizing aberrant splicing with one

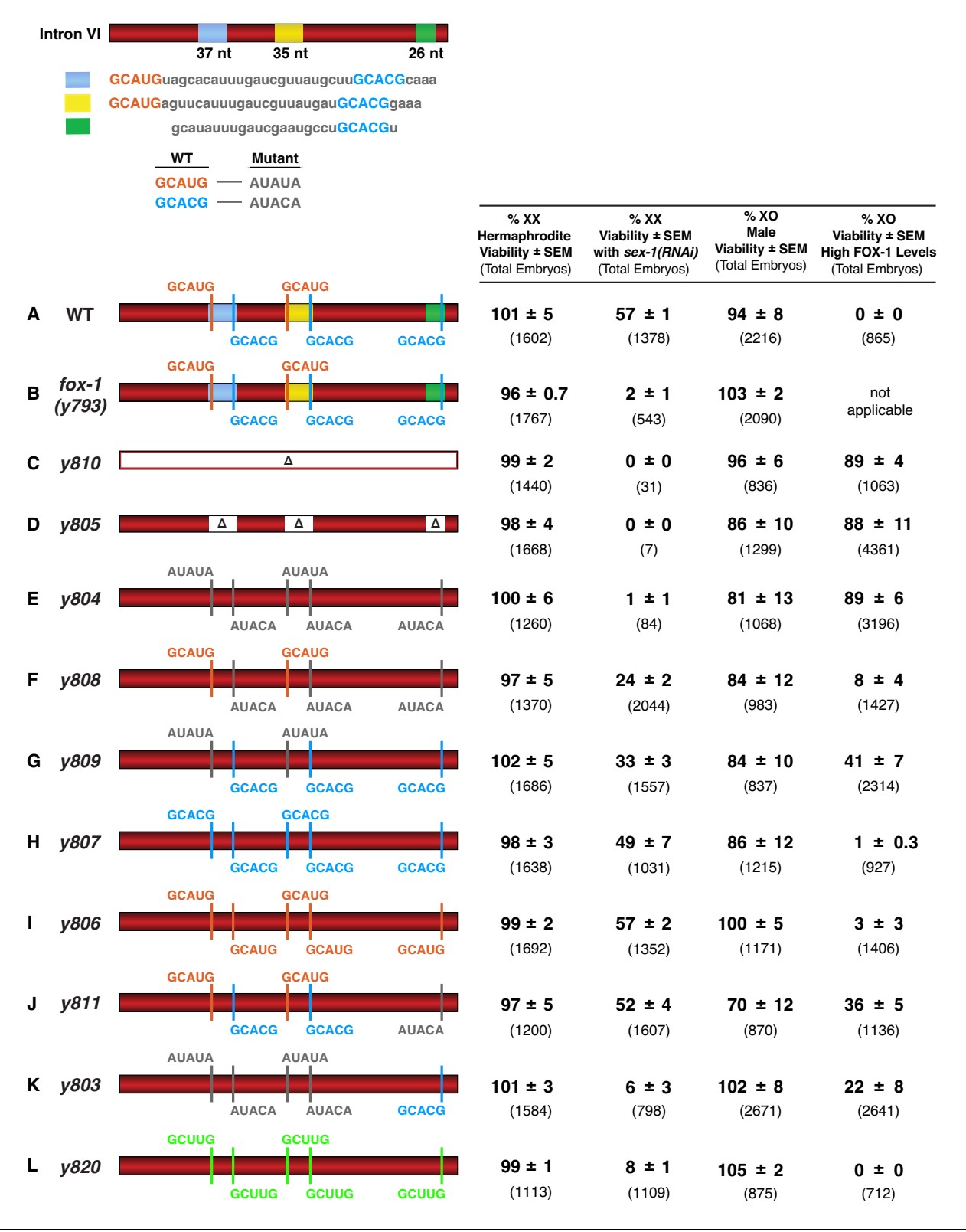

**Figure 6.** FOX-1 binds in vivo to multiple binding sites in *xol-1* intron VI using both GCAUG and GCACG motifs to regulate alternative splicing. The diagram of intron VI (top left) shows locations of the three regions (blue, yellow, and green) shown to exhibit FOX-1 binding in vitro. RNA sequences corresponding to each color-coded region are shown below the diagram. CRISPR/Cas9 editing was used to modify endogenous DNA encoding these regions and thereby identify *cis*-acting sites that control *xol-1* splicing in vivo. The motif GCAUG was changed to AUAUA, and the motif GCACU was

*Figure 6 continued on next page*

*Figure 6 continued*

changed to AUACA. (**A–L**) Diagrams of introns with the different CRISPR/Cas9 edits used for testing *xol-1* splicing regulation in vivo are shown on the left side. Multiple assays (right side) evaluate the effects on X:A signal activity of these intron mutations as well as a Cas9-induced deletion of endogenous *fox-1(y793)*. The effects on X:A signal activity are judged by the viability of XX and XO animals with different combinations of X-signal element (XSE) levels. Viability of XX hermaphrodites with mutations only in FOX-1 regulatory regions of *xol-1* measures the full contribution of splicing regulation toward X:A signal activity in the context of wild-type XSE levels and hence normal transcriptional repression by XSEs. Viability of XX *xol-1* mutants treated with RNAi against the XSE gene *sex-1*, which encodes a nuclear hormone receptor transcriptional repressor of *xol-1*, monitors the synergy between transcriptional and splicing regulation when transcriptional regulation is compromised such that *xol-1* expression is elevated. XX mutants were treated with *sex-1(RNAi)* for one generation to cause only partial *sex-1* inhibition and enable 57% survival. Viability of XO *xol-1* mutant animals in the context of high FOX-1 levels tests the efficacy of single and multiple FOX-1 binding sites on splicing regulation under conditions in which FOX-1 levels are not limiting for splicing regulation. The viability of *xol-1* XO mutant animals with wild-type FOX-1 levels serves as a control for any adverse effects of intron VI mutations unrelated to repression by excess FOX-1. All formulae for calculating viabilities of XX and XO animals with different XSE levels are provided in Materials and methods. For all assays, the average viability of multiple broods, each from a single hermaphrodite, is shown with the standard error of the mean (SEM). The total number of embryos scored for viability from all broods is indicated in parenthesis. The experiments show that splicing regulation becomes essential when transcriptional repression is compromised. Multiple FOX-1 binding sites utilizing GCACG and GCAUG motifs are required for full splicing regulation. The number of binding sites is more critical than whether the motif sequence is GCAUG or GCACG. However, replacing all high-affinity motifs with the low-affinity secondary motif GCUUG promotes only minimal non-productive alternative splicing. High-affinity motifs are essential for FOX-1 mediated repression. Eliminating intron VI revealed no greater benefits for male viability than deleting the entire *fox-1* gene.

The online version of this article includes the following source data and figure supplement(s) for figure 6:

**Source data 1.** Overexpression of ASD-1 kills both XO males and XX hermaphrodites.

**Figure supplement 1.** Brood sizes of XX animals with *cis*-acting *xol-1* mutations in intron VI are consistent with the strain viability.

**Figure supplement 2.** High FOX-1 levels in XX animals repress *xol-1* with multiple low-affinity GCUUG motifs in intron VI.

dose of *fox-1* to allow high *xol-1* activity for male development and by increasing aberrant splicing with two doses of *fox-1* to allow low *xol-1* activity for hermaphrodite development.

We used multiple assays to judge the impact on *xol-1* splicing regulation caused by disrupting endogenous FOX-1 binding sites. First, we assessed the viability of XX hermaphrodites carrying *xol-1* mutations in FOX-1 regulatory regions. This assay measures the contribution of *xol-1* splicing regulation toward X:A signal assessment in the context of full transcriptional repression by other X signal elements, SEX-1 (nuclear hormone receptor) and CEH-39 (homeodomain protein). Second, we assessed the viability of XX *xol-1* mutant hermaphrodites in the context of reduced SEX-1 activity, and hence elevated *xol-1* transcription, to measure synergy between transcriptional and splicing regulation. This sensitized XSE mutant condition was achieved using RNA interference (RNAi) against *sex-1*. Third, we assessed the viability of XO *xol-1* mutant males in the context of high FOX-1 levels that are sufficient to kill all otherwise wild-type XO animals by causing non-productive alternative splicing. In these XO animals, the single dose of *sex-1* and *ceh-39* does not repress *xol-1* expression. This sensitive assay measures the efficacy of single and multiple wild-type FOX-1 binding sites on splicing regulation under conditions in which FOX-1 levels are not limiting.

In initial experiments, we eliminated the non-productive alternative splicing mode of *xol-1* repression using CRISPR/Cas9 editing to fuse exon 6 in frame with exon 7 at the endogenous *xol-1* locus (*y810*) and thereby exclude intron VI from the pre-mRNA. In XO animals, removing intron VI blocked the XO-specific lethality caused by overexpressing FOX-1. Viability of XO males increased from 0% to 89% ($p<10^{-5}$) for *y810*, indicating that splicing regulation was severely disrupted, as predicted (***Figure 6A,C***).

In XX animals, loss of intron VI did not reduce either the viability of mutant (*y810*) versus wild-type animals (99% vs. 101%, respectively) or the average brood size per hermaphrodite (240 ± 4 vs. 267 ± 32 embryos) (***Figure 6—figure supplement 1A,C***). Consistent with this intron VI deletion result, a null mutation of *fox-1(y793)* created by a Cas9-induced deletion of the entire endogenous gene also resulted in insignificant XX lethality and no reduction in brood size (***Figure 6B*** and ***Figure 6—figure supplement 1B***). However, just as eliminating FOX-1 activity by gene deletion killed XX hermaphrodites sensitized by reduced activity of the XSE transcription repressor SEX-1, eliminating *xol-1* intron VI killed all XX hermaphrodites with reduced SEX-1 activity (***Figure 6C***). XX viability decreased from 57% to 2% ($p<10^{-5}$) for deletion of *fox-1(y793)* and from 57% to 0% ($p<10^{-5}$) for deletion of the intron (*y810)* (***Figure 6C*** and ***Figure 6—figure supplement 1C***). Thus, repression of

*xol-1* by splicing regulation in XX animals becomes critical primarily in the context of compromised transcriptional repression.

One other FOX-1 family member, the autosomal protein ASD-1, binds GCAUG motifs and controls alternative splicing of other *C. elegans* developmental regulators (*Kuroyanagi et al., 2013*; *Kuroyanagi et al., 2007*; *Kuroyanagi et al., 2006*). Our genetic evidence suggested it does not regulate *xol-1* either in the presence or absence of FOX-1 (*Figure 6—source data 1*). That possibility could not have been fully eliminated until intron VI mutations removed the opportunity for any RNA binding factors to regulate *xol-1* alternative splicing. No greater benefit to XO males or greater detriment to XX hermaphrodites occurred when all potential sources of splicing regulation were eliminated by removing intron VI rather than by deleting *fox-1* alone.

## FOX-1 binding sites identified in vitro function in vivo to regulate *xol-1* splicing

To determine whether FOX-1 binding sites identified in vitro by biochemical analysis do indeed function in vivo to regulate *xol-1* RNA splicing, we used two strategies to mutate endogenous DNA encoding these sites in intron VI. First, we deleted the 37 bp, 35 bp, and 26 bp regions corresponding to the in vitro FOX-1 binding sites (*y805*) (*Figure 6D*). Second, we altered the sequence of all individual putative binding motifs within each binding site (*y804*) (*Figure 6E*). Endogenous DNA was edited to convert the RNA motif GCAUG to AUAUA and GCACG to AUACA.

The nucleotide deletions (*y805*) and substitutions (*y804*) of all FOX-1 binding sites and motifs, respectively, within intron VI had the same effect as eliminating the intron: nearly complete loss of XX viability in the sensitized *sex-1(RNAi)* XSE mutant condition and suppression of male lethality caused by FOX-1 overexpression (*Figure 6D,E*). XX viability with *sex-1(RNAi)* was reduced from 57% to 0% ($p<10^{-5}$) by the binding site deletions, and from 57% to 1% ($p<10^{-5}$) by the motif substitutions. Male viability with high FOX-1 levels increased from 0% to 88% ($p<10^{-5}$) with the deletions and from 0% to 89% ($p<10^{-5}$) with the substitutions. In contrast, the viability and brood sizes of XX animals bearing only deletions or nucleotide substitutions of all FOX-1 binding sites and motifs in a *sex-1*(+) condition were not different from wild-type XX animals (*Figure 6D,E* and *Figure 6—figure supplement 1D,E*). These experiments indicate that FOX-1 binding sites and motifs identified in vitro do function in vivo to mediate *xol-1* splicing repression, and confirm that splicing regulation is essential for hermaphrodite viability primarily in the context of reduced transcriptional repression.

## Multiple GCAUG motifs and GCACG motifs in intron VI are essential for FOX-1-regulated alternative *xol-1* splicing

Genome editing of *fox-1* binding sites also allowed us to determine the efficacy in vivo of GCAUG motifs versus GCACG motifs in splicing-mediated *xol-1* repression in XX and XO animals and to determine how many FOX-1 binding sites are required for full splicing regulation. Mutating the three GCACG motifs while retaining the two GCAUG motifs (*y808*) reduced the viability of *sex-1(RNAi)* XX animals to about half the level of *sex-1(RNAi)* XX animals with five wild-type motifs (24% vs. 57%) ($p<10^{-5}$), but permitted more XX viability than with mutant versions of all five GCAUG and GCACG motifs (*y804*) (24% vs. 1%) ($p<10^{-5}$) (*Figure 6F* and *Figure 6—figure supplement 1F*). These results indicate that GCACG motifs function in vivo for splicing repression in XX animals, but by themselves are not sufficient for full repression; the GCAUG motifs are also required. The two GCAUG motifs remaining in *y808* also severely reduced the viability of XO animals with high FOX-1 levels compared to those with five mutant motifs (*y804*) (8% vs. 89%) ($p<10^{-5}$), demonstrating that GCAUG motifs contribute to FOX-1-mediated repression in XO animals (*Figure 6F*).

Reciprocally, mutating the two GCAUG motifs while retaining the three wild-type GCACG motifs (*y809*) reduced the viability of XSE-sensitized XX animals by about half compared to those sensitized XX animals having all five wild-type motifs (33% vs. 57%) ($p<10^{-4}$), but permitted more XX viability than all five mutant motifs (*y804*) (33% vs. 1%) ($p<10^{-5}$) (*Figure 6G* and *Figure 6—figure supplement 1G*). These results indicate that GCAUG motifs function in vivo for splicing repression in XX animals but are not sufficient for full repression; GCACG motifs are also important. The three GCACG motifs remaining in *y809* reduced the viability of XO animals with high FOX-1 levels by half compared to XO animals with all mutant motifs (*y804*) (41% vs. 89%) ($p<10^{-5}$) (*Figure 6G*), indicating that GCACG motifs contribute to splicing regulation in XO animals. Thus, multiple FOX-1

binding sites are required in vivo for *xol-1* repression by alternative splicing, and both GCACG and GCAUG motifs are important.

To determine whether GCACG motifs alone or GCAUG motifs alone could be sufficient for full splicing-mediated repression when present at all five sites, we converted all five motifs to either GCACG (*y807*) or GCAUG (*y806*) (*Figure 6H,I* and *Figure 6—figure supplement 1H,I*). When present in all sites, either GCACG or GCAUG motifs were sufficient to confer complete splicing repression, just like a wild-type intron VI. Virtually all XO animals were killed by high levels of FOX-1 (1% or 3%, respectively, vs. 0%), and the viability of XSE-sensitized XX animals was equivalent to that achieved with a wild-type intron VI (49% or 57%, respectively, vs. 57%) (*Figure 6H,I*). These results indicate that the number of binding sites is more important than whether the sequences are GCAUG or GCACG.

The significance of motif number in splicing repression is further illustrated by additional experiments. First, XX viability was compared directly between edited animals that have only GCAUG motifs but differ in GCAUG number. Viability of *sex-1(RNAi)* XX animals with five GCAUG motifs (*y806*) was 57%, but viability with two GCAUG motifs was 24% ($p < 10^{-4}$) (*Figure 6I* and *Figure 6—figure supplement 1I*). Second, viability of *sex-1(RNAi)* XX animals was relatively normal with four of five wild-type motifs (*y811*) (52% vs. 57%) (*Figure 6J* and *Figure 6—figure supplement 1J*). However, viability of *sex-1(RNAi)* XX animals was reduced when *xol-1* had just two or three wild-type motifs: 24% ($p < 10^{-5}$) for two GCAUG motifs in *y808%* and 33% ($p < 10^{-4}$) for three GCACG motifs in *y809* (*Figure 6F,G*). Third, mutation of a single motif, the terminal GCACG motif (*y811*), was sufficient to reduce splicing-mediated repression of *xol-1* in XO animals by high FOX-1 levels (*Figure 6J*). More XO animals with excess FOX-1 were viable with only four of five wild-type motifs (*y811*) than with all five (36% vs. 0%). Fourth, in a reciprocal experiment, a single GCACG motif (*y803*) was sufficient to decrease viability of XO animals with high FOX-1 levels compared to XO animals with no wild-type sites (*y804*) (22% vs. 89%) ($p < 10^{-5}$) (*Figure 6K*). Thus, in the context of high FOX-1 levels, a single GCACG motif functions in splicing repression, but increasing the number of either GCACG or GCAUG motifs causes progressively greater repression and less XO viability.

To further assess the effect of only a single FOX-1 binding site in *xol-1*, we examined XX animals in which only the terminal GCACG motif (*y803*) was present (*Figure 6K* and *Figure 6—figure supplement 1K*). We found severe XX lethality in the XSE-sensitized background (6% viable), indicating that the single GCACG site by itself is not sufficient for robust regulation in XX animals. Multiple sites are required, as exemplified from the increased viability of *sex-1(RNAi)* XX animals with three GCACG motifs (33%, $p < 10^{-5}$) (*Figure 6G*) and five GCACG motifs (49%, p=0.01) (*Figure 6H*).

The fact that 6% of XX animals were viable with just the single site (*y803*) instead of 1% (p=0.002) when all sites were altered (*y804*) indicates that the GCACG site by itself permits sufficient alternative splicing to rescue some XX animals (*Figure 6E,K*). Utility of the single site is also reflected in the brood size of these XSE-sensitized animals. Hermaphrodites with one GCACG site had an average brood size of 100 ± 25 embryos, while those with no sites had an average brood size of 11 ± 4 embryos (*Figure 6—figure supplement 1E,K*). Consistent with the efficacy of a single GCACG site, viability of XO animals in the presence of high FOX-1 levels was reduced from 89% when all sites were absent (*y804*) to only 22% ($p < 10^{-5}$) when a single GCACG site (*y803*) was present (*Figure 6E, K*). Thus, while one binding site can achieve some repression, multiple sites are needed for full repression, thereby creating a sensitive mechanism for *xol-1* regulation by FOX-1.

No GCAUG or GCACG motifs are present in any *xol-1* intron other than intron VI, consistent with intron VI being sufficient for FOX-1 regulation. Two GCACG motifs are present in the 3' UTR of the 2.2 kb transcript, but this 3' UTR is not necessary for repression by FOX-1 (*Figure 4B*).

## Multiple low-affinity GCUUG motifs in intron VI are not sufficient to repress *xol-1* with the FOX-1 levels present in wild-type XX or XO animals but are sufficient for *xol-1* repression in both sexes with high FOX-1 levels

Recent studies demonstrated that mammalian Rbfox can utilize secondary, low-affinity binding motifs such as GCUUG to promote alternative splicing in vivo if these motifs are present in multiple copies, and if Rbfox reaches a concentration higher than necessary for binding to GCAUG motifs (*Begg et al., 2020*). Therefore, we tested whether replacing all GCACG and GCAUG motifs in intron VI with GCUUG motifs would promote sufficient non-productive splicing in the presence of high

FOX-1 levels to cause the death of XO males and, reciprocally, the survival of XX embryos exposed to *sex-1(RNAi)*. In XO animals with high FOX-1 levels, the low-affinity GCUUG binding sites were sufficient to cause complete XO lethality (*y820* in *Figure 6L*).

In XX animals with wild-type GCUCG and GCACG motifs in intron VI, high levels of FOX-1 could suppress the death caused by *sex-1(RNAi)* (24% vs. 63% viable p=0.004) (*Figure 6—figure supplement 2A,B*), demonstrating that enhancing post-transcriptional repression of *xol-1* can compensate for the deficiency in repression caused by reducing transcriptional repression. Although high FOX-1 levels in XX animals with mutated GCACG and GCAUG motifs could not suppress any death caused by *sex-1(RNAi)* (*Figure 6—figure supplement 2D*), high FOX-1 levels could rescue XX animals with low-affinity GCUUG motifs from *sex-1(RNAi)*-induced death (p=0.006) (*Figure 6—figure supplement 2C*), as could high FOX-1 levels with normal GCAUG and GCACG motifs.

In contrast, GCUUG motifs were not adequate to suppress the lethality caused by *sex-1(RNAi)* when only the two wild-type doses of *fox-1* were present in XX embryos. Only 8% of *y820 sex-1 (RNAi)* XX animals carrying GCUUG motifs survived compared to 57% survival when all sites had the original high-affinity GCAUG or GCACG motifs (p<10$^{-3}$) (*Figure 6L* and *Figure 6—figure supplement 1L*). These results show that although multiple low-affinity GCUUG binding sites, coupled with high FOX-1 levels, are sufficient to kill XO animals or to suppress the death of *sex-1(RNAi)* XX animals, the low-affinity GCUUG sites are inadequate to repress *xol-1* in XX embryos with only the normal two doses of *fox-1*. The degree of non-productive splicing necessary to repress *xol-1* in XX embryos with two copies of *fox-1* is only reached if multiple high-affinity motifs are present in intron VI.

## FOX-1 acts in a dose-dependent manner in XX animals to regulate *xol-1* splicing and thereby determines sex

The need for multiple high-affinity binding sites to enable two doses of FOX-1 to repress *xol-1* in XX animals suggested that *xol-1* splicing control would be sensitive to FOX-1 dose. We took two approaches to evaluate the dose-dependence of FOX-1 action in determining sex. In XX animals with reduced *sex-1* activity caused by RNAi directed against *sex-1*, we compared the impact on *xol-1* regulation of reducing the dose of FOX-1 from two copies to one to the impact of mutating combinations of FOX-1 binding sites in one endogenous copy of *xol-1*. Both approaches revealed dose-sensitivity of FOX-1 action.

Viability of the *sex-1(RNAi)* XX animals declined more than 10-fold, from 44% to 3% (p=0.002), when the dose of *fox-1* was reduced by half, from two copies to one copy, demonstrating dose-dependence of FOX-1 function in XX animals (*Figure 7A,B*). Similarly, viability of *sex-1(RNAi)* XX animals declined from 44% to 7% (p=0.006) when all FOX-1 binding sites in intron VI were mutated in one copy of *xol-1* (*Figure 7A,C*). As expected, viability of *sex-1(RNAi)* XX animals declined to an intermediate level when either the three GCACG motifs (18%) (p=0.018) or the two GCAUG motifs (13%) (p=0.012) were mutated in one copy of *xol-1* (*Figure 7A,D,E*). Viability of *sex-1(RNAi)* XX animals with five low-affinity GCUUG motifs in one copy of *xol-1* was better (36%) than that with either two GCAUG (p=0.023) or three GCACG (p=0.017) mutated motifs in one *xol-1* copy (*Figure 7F*). Hence, the dose-dependence of FOX-1 function in regulating alternative *xol-1* splicing in XX animals is evident both from reducing the dose of the *trans*-acting FOX-1 protein or by mutating different combinations of *cis*-acting FOX-1 binding sites in only one copy of *xol-1*. Thus, FOX-1 acts as an XSE to convey X-chromosome number by regulating alternative *xol-1* splicing in a dose-dependent manner.

## Discussion

We dissected the mechanism by which the *C. elegans* RNA binding protein FOX-1 acts as a dose-dependent X-chromosome signal element to specify sexual fate. In XX embryos, FOX-1 binds to the single alternatively spliced intron of *xol-1*, the master regulator that sets the male fate in XO embryos, and causes either intron retention, and hence premature translation termination, or alternative 3' acceptor site usage, and hence exclusion of essential exon coding sequences (*Figure 8*). Both events prevent production in XX embryos of functional male-determining XOL-1 protein, which also sets the level of X-chromosome gene expression in XO embryos. FOX-1 must bind to multiple high-affinity GCAUG and GCACG motifs in *xol-1* intronic sequences to regulate *xol-1* splicing in XX

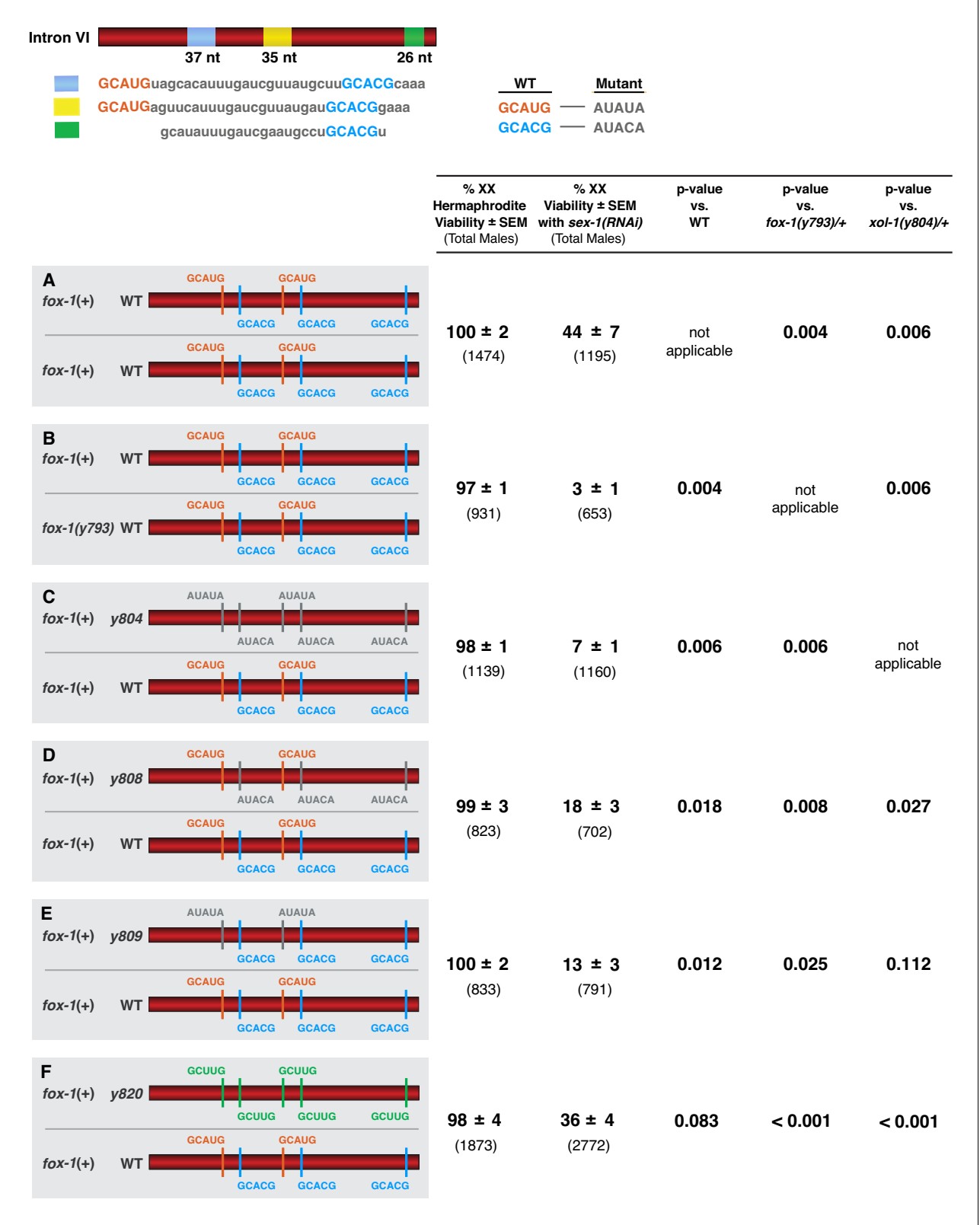

**Figure 7.** FOX-1 acts in a dose-dependent manner to regulate *xol-1* splicing in XX animals and thereby determine sex. (A–F) Diagrams on the left show sequences for the two different *xol-1* intron VI combinations assayed in each cohort of *sex-1(RNAi)* XX animals to assess the dose-dependence of FOX-1 action in regulating *xol-1* splicing. Viability of both *sex-1(+)* and *sex-1(RNAi)* animals is shown on the right along with statistical comparisons of viability across different genotypes. Except for (A, B) in which both copies of intron VI have unaltered FOX-1 binding sites, one *xol-1* intron VI has

*Figure 7 continued on next page*

*Figure 7 continued*

mutated FOX-1 binding sites as indicated and one intron has unaltered FOX-1 binding sequences (**C–F**). (**B**) Low viability of *sex-1(RNAi)* XX animals with only one dose of *fox-1* [*fox-1(y793)* / +] shows strong dose-dependence of FOX-1 action in *xol-1* splicing regulation. (**C–F**) The impact of heterozygous combinations of intron VI mutations on viability of *sex-1(RNAi)* XX animals also indicates that FOX-1 functions as a dose-dependent X signal element to regulate *xol-1* splicing and thereby communicate X-chromosome dose. Viability assays were conducted using the protocols that follow. Separate matings were performed between *Prps*-0::mNeonGreen::4xNLS::unc-54 green males and hermaphrodites of genotypes: wild-type XX for (**A**), *fox-1(y793)* XX for (**B**), *xol-1(y804)* XX for (**C**), *xol-1(y808)* for (**D**), *xol-1(y809)* for (**E**), and *xol-1(y820)* for (**F**). For the *sex-1(RNAi)* experiments, matings were performed on plates with bacteria containing plasmids that produce double-stranded *sex-1* RNA when XX animals were young adults. For the control set of matings, males and hermaphrodites were grown on bacteria that do not produce double-stranded *sex-1* RNA (pL4440 empty vector control). For both sets of crosses, all green hermaphrodites and green males were counted. Since viability of XO animals is not affected by *sex-1(RNAi)*, the number of green XX hermaphrodites expected if all were viable would be the same as the number of green XO males. Percent XX viability was calculated by (Number of green hermaphrodites/Number of green males) $\times$ 100.

embryos, and this splicing regulation is dose-dependent to achieve *xol-1* repression in XX but not XO embryos. Deleting one copy of *fox-1* or removing all FOX-1 binding sites in one copy of *xol-1* reduces splicing regulation sufficiently to kill XX animals with reduced XSE activity, demonstrating the importance of *fox-1* dose for viability of XX animals. Having two doses of *fox-1* in XX embryos is as important for viability as restricting the dose of *fox-1* to one in XO embryos.

Dose-dependent regulation of *xol-1* splicing acts as a secondary mode of repression in XX embryos to enhance the fidelity of X:A signaling. Transcriptional repression by XSEs is the primary mode of *xol-1* regulation (*Carmi et al., 1998*; *Carmi and Meyer, 1999*; *Farboud et al., 2013*; *Gladden and Meyer, 2007*; *Meyer, 2018*; *Powell et al., 2005*). We showed that non-productive alternative splicing is then imposed on residual *xol-1* transcripts to achieve full *xol-1* repression in XX embryos. Repression of *xol-1* by splicing regulation is especially critical in the context of compromised transcriptional repression and during early development prior to maximal transcriptional repression. XX embryos lacking splicing regulation die if *xol-1* transcription is even partially activated in XX embryos. Reciprocally, increasing FOX-1 concentration above the normal level in XX embryos, and hence increasing non-productive splicing, suppresses the XX lethality caused by reducing *xol-1* transcriptional repression. The combined action of splicing and transcriptional repression of *xol-1* enhances the precision of X-chromosome counting.

## Mechanisms underlying the action of FOX family members in regulating alternative pre-mRNA splicing

Multiple high-affinity FOX-1 binding motifs are necessary to restrict the non-productive mode *xol-1* splicing to XX embryos over the small FOX-1 concentration range that distinguishes XX from XO embryos. Among numerous experiments, the need for multiple high-affinity RNA binding sites to repress *xol-1* was well exemplified by the comparison in XX viability of engineered *xol-1* strains that carried only GCACG motifs in intron VI, but in different numbers, and were subjected to *sex-1 (RNAi)*. The viability of the XX strain with five GCACG motifs (49%) was greater than that with three GCACG motifs (33%) or with one motif (6%). Moreover, replacing all GCUAG and GCACG motifs with low-affinity GCUUG motifs permitted only minimal survival (8% vs. 57%).

In other cases of splicing regulation by *C. elegans* FOX-1, specifically the *unc-32* and *egl-15* gene targets, robust regulation is achieved through a single GCAUG binding site using a different strategy from the one for *xol-1*. FOX-1 binds the *unc-32* pre-mRNA at the single GCAUG site in concert with the neuronally expressed CELF RNA binding protein UNC-75 to promote skipping of the upstream exon (*Kuroyanagi et al., 2013*). To do so, FOX-1 acts redundantly with ASD-1, another FOX-1 family member. Either FOX-1 or ASD-1 can bind the GCAUG site with UNC-75, and together regulate *unc-32*. Similarly, FOX-1 and ASD-1 regulate splicing of *egl-15* pre-mRNA in combination with the muscle-specific RNA binding protein SUP-12 using a single GCAUG site (*Kuroyanagi et al., 2007*). For both gene targets, two FOX-1 family members bind a single binding site to help ensure proper splicing in conjunction with key tissue-specific RNA binding proteins. Loss of either *fox-1* or *asd-1* activity causes partial loss-of-function phenotypes for *egl-15* and for *unc-32*, revealing that *asd-1* and *fox-1* function in a collaborative fashion to reinforce splicing control of two different pre-mRNA targets, each with one binding site (*Kuroyanagi et al., 2013*; *Kuroyanagi et al., 2007*). In contrast, our

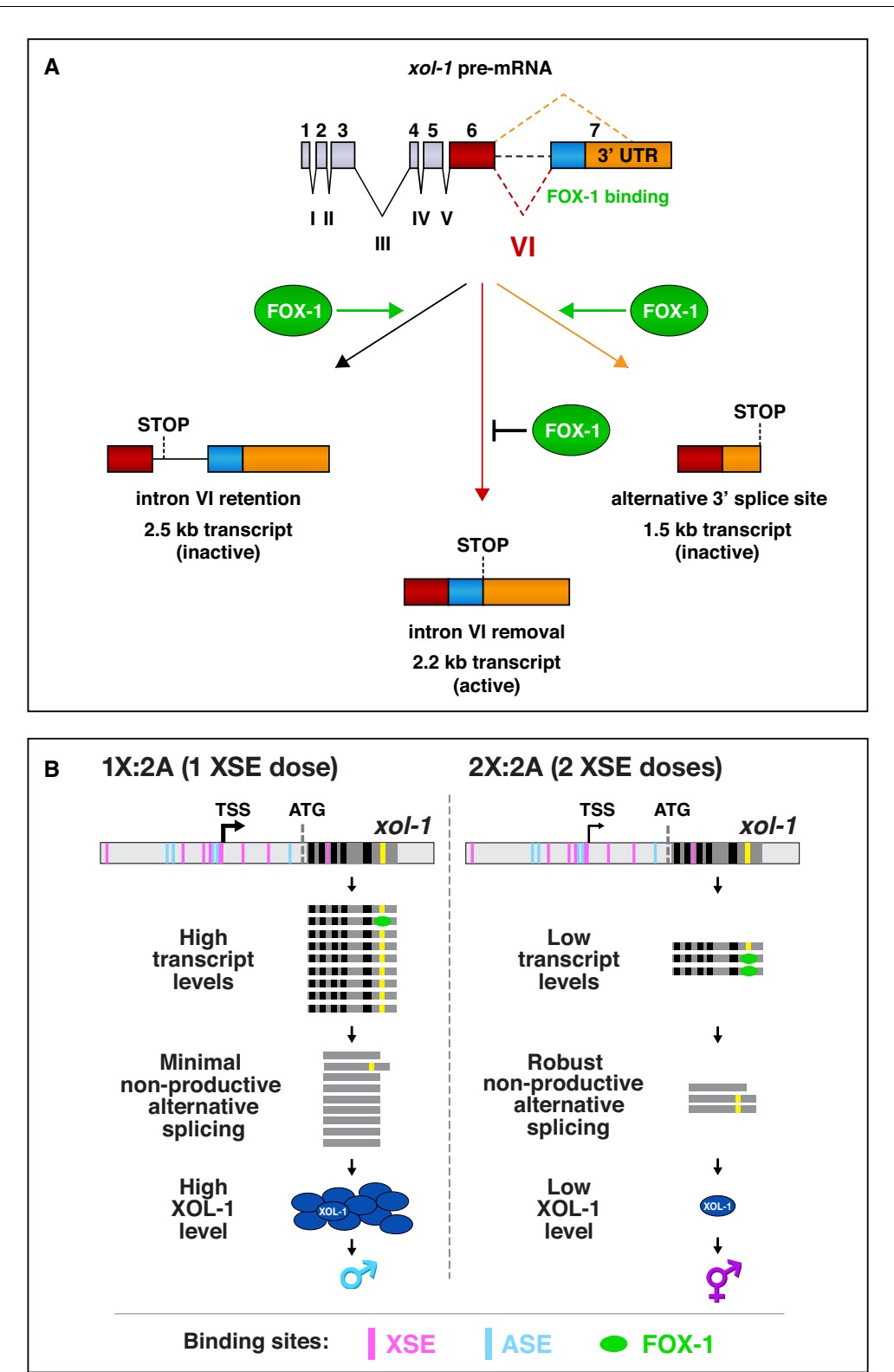

**Figure 8.** Summary of *xol-1* splicing regulation by FOX-1 and model for X:A signal assessment. (**A**) Summary of *xol-1* splicing regulation by FOX-1. Through binding to multiple GCAUG and GCACG motifs in intron VI of *xol-1*, FOX-1 reduces formation of the male-determining 2.2 kb transcript by causing intron VI retention (2.5 kb transcript) or by directing use of an alternative 3' splice acceptor site, causing deletion of essential exon 7 coding sequences (blue) and part of the 3' UTR (orange) (1.5 kb transcript). (**B**) Model for X:A signal assessment: two tiers of *xol-1* repression. X-signal

*Figure 8 continued*

elements (XSEs) and autosomal signal elements (ASEs) bind directly to numerous non-overlapping sites in the 5' regulatory region of *xol-1* to antagonize each other's opposing transcriptional activities and thereby control *xol-1* transcription (*Farboud et al., 2013*). Molecular rivalry at the *xol-1* promoter between the XSE transcriptional repressors and ASE transcriptional activators causes high *xol-1* transcript levels in 1X:2A embryos with one dose of XSEs and low levels in 2X:2A embryos with two doses of XSE. All binding sites for the XSEs (nuclear receptor SEX-1 and homeodomain protein CEH-39) are shown in magenta and binding sites for the T-box transcription factor ASE called SEA-1 are shown in blue. Binding sites for the zinc finger ASE called SEA-2 have not been mapped precisely enough in this *xol-1* regulatory region to represent. In a second tier of *xol-1* repression shown by our studies, the XSE RNA binding protein FOX-1 (green) then enhances the fidelity of X-chromosome counting by binding to numerous GCAUG and GCAUG motifs in intron VI (yellow) of the residual *xol-1* pre-mRNA, thereby causing non-productive alternative splicing and hence *xol-1* mRNA variants that have in-frame stop codons or lack essential exons. High XOL-1 protein induces the male fate and low XOL-1 permits the hermaphrodite fate. Black rectangles represent *xol-1* exons, dark gray rectangles represent *xol-1* introns, and light gray rectangles represent 5' and 3' *xol-1* regulatory regions.

results showed that ASD-1 is not required for regulation of alternative *xol-1* splicing to repress *xol-1* activity in XX embryos.

For many mammalian genes, Rbfox can bind to an intron and control pre-mRNA splicing using a single high-affinity GCAUG motif. A tyrosine-rich, low-complexity domain at the C-terminus then nucleates aggregation of Rbfox to attain the concentration of bound proteins necessary to drive alternative splicing (*Ying et al., 2017*). Rbfox aggregation is essential for inclusion of specific exons into mature mRNA and has the potential to recruit additional splicing factors. Rbfox aggregation may also be necessary for concentrating Rbfox complexes in nuclear speckles, where RNA synthesis occurs (*Ying et al., 2017*). Such concentrated localization may promote binding to and alternative splicing of newly synthesized RNAs. *C. elegans* FOX-1 lacks a tyrosine-rich low-complexity domain to cause protein aggregation, but during *xol-1* regulation, recruitment of FOX-1 through multiple intronic binding sites using GCAUG and GCACG motifs appears to substitute for protein aggregation in achieving robust alternative splicing. In this case, availability of multiple sites would increase the probability of any site being occupied.

Genome-wide binding studies of Rbfox revealed that many mammalian introns have more than one primary GCAUG or GCACG motif (*Begg et al., 2020*; *Lambert et al., 2014*). While experiments have not yet addressed whether the multiple motifs enhance splicing regulation at these endogenous sites, other experiments using mammalian reporter constructs have demonstrated that three GCAUG motifs are more effective at regulating splicing than a single motif (*Modafferi and Black, 1999*).

The importance of multiple motifs in mammalian pre-mRNA splicing regulation is further exemplified by a recent study showing that a cluster of six Rbfox low-affinity secondary binding motifs, such as GCUUG or GUUUG, but not a single low-affinity motif, controls splicing regulation during neuronal differentiation in vivo, as the Rbfox concentration naturally increases during development (*Begg et al., 2020*). Higher Rbfox concentrations are required for splicing regulation using the clustered low-affinity sites than those required for splicing regulation by a single high-affinity GCAUG motif. Hence, the clustered motifs create tissue and temporal specificity in splicing regulation in response to changes in Rbfox concentration during development (*Begg et al., 2020*).

A cluster of intronic *xol-1* motifs could tune splicing regulation over the twofold increase in FOX-1 concentration between XO and XX embryos, as Rbfox appears to do during neuronal differentiation (*Begg et al., 2020*). The cluster of *xol-1* motifs likely enhances the probability that the higher FOX-1 concentration in XX embryos versus XO embryos reaches the threshold necessary for FOX-1 binding and function, thereby restricting splicing regulation to XX embryos. Our studies showed that a single FOX-1 binding site becomes sufficient to induce some non-productive *xol-1* splicing in XO embryos and thereby kill them when the concentration of FOX-1 is increased artificially by integrating numerous extra copies of *fox-1* into the genome. However, multiple high-affinity motifs are necessary to achieve the non-productive mode of *xol-1* splicing to XX embryos. Even for *xol-1*, though, secondary GCUUG motifs can achieve splicing regulation if the FOX-1 concentration is elevated beyond the normal level in XX embryos. Five GCUUG motifs are sufficient to permit high FOX-1 levels to suppress the XX-specific lethality caused by *sex-1(RNAi)* or to kill all XO embryos. These results converge with and extend those found for mammalian Rbfox.

## Strategies to determine sex: Comparison across species of primary sex-determination mechanisms requiring both transcriptional and pre-mRNA splicing regulation

Like *C. elegans*, the fruit fly *D. melanogaster* utilizes the combination of transcriptional and pre-mRNA splicing regulation to enhance the precision of X:A counting by controlling *Sxl* (*Sex-lethal*), the master sex-determination switch gene and direct target of the X:A signal. This sex switch gene dictates female development when active and permits male development when inactive (*Cline and Meyer, 1996*). *Sxl* encodes an X-linked RNA binding protein with an RRM. It directs female development by regulating alternative pre-mRNA splicing of itself and downstream sex-determination genes. *Sxl* is activated in 2X:2A embryos, but not 1X:2A embryos, by a set of feminizing XSEs that stimulate transcription of *Sxl* in a dose-dependent manner. Once produced, SXL protein functions in a positive autoregulatory loop to control splicing of its own pre-mRNA in a dose-dependent manner and thereby promote continued production of female-specific SXL protein (*Bell et al., 1991*; *Cline and Meyer, 1996*; *Horabin and Schedl, 1993*; *Sakamoto et al., 1992*). By binding to two neighboring *Sxl* introns, SXL prevents inclusion of the intervening male-specific exon 3, which encodes an in-frame stop codon that prevents translation of the full-length female SXL protein when incorporated into mature RNA.

SXL also directs pre-mRNA splicing of its downstream sex-determination target gene called *tra* (transformer), a switch gene that directs female sexual differentiation (*Inoue et al., 1990*; *Sosnowski et al., 1989*; *Valcárcel et al., 1993*). SXL controls *tra* by regulating 3′ splice site selection. By binding to an upstream 3′ splice site in the first intron of *tra* pre-mRNA, SXL competes with binding of splicing factor U2AF and thereby diverts splicing to a distal 3′ splice acceptor site in that intron. The resulting shorter RNA isoform encodes a full-length TRA protein.

Thus, SXL dictates female development by directing formation of RNA isoforms for itself and downstream targets that encode essential female-specific proteins. In contrast, FOX-1 promotes hermaphrodite development by dose-dependent binding to intron VI of the male-determining gene *xol-1*, causing either intron retention or deletion of exon coding sequences, and thereby blocking formation of the RNA variant that encodes male-specific XOL-1 protein. While *fox-1* acts as a worm XSE to communicate X-chromosome dose, *Sxl* responds to fly XSEs to determine sex, but its location on X and its autoregulatory feature allow it to serve as both signal and target.

## Materials and methods

### Production and assay of *xol-1* transgenes in extrachromosomal arrays

The parental transgene plasmid (pMN45) that *Nicoll et al., 1997* used to construct deletion derivatives diagrammed in *Figure 2A* is a genomic *xol-1* rescuing clone that has *gfp* coding sequences inserted at the first ATG of *xol-1.* In extra-chromosomal arrays made by injecting 5 μg/ml plasmid DNA, the multi-copy transgenes produce a bifunctional GFP::XOL-1 protein that rescues the lethality of *xol-1* XO null mutants and expresses GFP in a sex-specific manner, high in XO and low in XX embryos. Although arrays of *xol-1*(+) transgenes from 5 μg/ml injections exhibited sex-specific regulation, the levels of XOL-1 were higher in XX animals than those produced by two copies of the endogenous gene. This elevated level of *xol-1* expression made XX animals dependent on endogenous FOX-1 activity to repress the transgenes and achieve full XX viability. Because *xol-1*(+) transgenic arrays cause XX lethality when *fox-1* is inactive, reporters with deletion derivations that eliminate nucleotides required for FOX-1 regulation are expected to cause XX lethality or at least milder dosage compensation phenotypes such as a dumpy (Dpy) body shape and an egg-laying defect (Egl). Prior experiments showed that excess FOX-1 is unable to repress a translational *xol-1::gfp* reporter that lacks sequences downstream of the first two introns and 89 codons of *xol-1*, allowing us to keep intron I in the reporters. Because we found that the phenotypic analysis of XOL-1 activity (Dead, Viable, Dpy Egl) was a much more sensitive indicator of *xol-1* activity than GFP fluorescence, we reported only the phenotypic analysis. As an example of the difference in sensitivity, we found that derepression of the *gfp::xol-1* transgene caused XX-specific lethality and/or a Dpy phenotype even for cases in which GFP fluorescence did not appear to increase as measured by a fluorescence dissecting microscope.

Arrays of *xol-1*(+) transgenes could only be established using a *xol-1*(+) DNA injection concentration at 5 µg/ml. Any higher *xol-1*(+) DNA injection concentration killed XX animals, and extra-chromosomal lines could not be established. All deletion derivatives of *xol-1* transgenes were also injected with a DNA concentration of 5 µg/ml. The *xol-1* plasmids were co-injected with 150 µg/ml of *unc-76*(+) marker plasmid p76-16b into *him-5(e1490) unc-76(e911); xol-1(y9)* hermaphrodites. *him-5 unc-76; xol-1* hermaphrodites produce 67% viable Unc XX hermaphrodites and 33% Unc XO males that die as embryos or L1 larvae in the absence of *xol-1*(+) from a transgenic array. Extra-chromosomal arrays have variable transmission through meiosis and can only be followed when the arrays carry DNA encoding a wild-type genetic maker, such as *unc-76*(+), and they are established in animals carrying an *unc-76* mutation, which causes the animals to be Unc. An array resulting in only Unc hermaphrodites and both Unc and non-Unc males indicate the transgenic array causes XX-specific lethality. Lines were maintained through non-Unc animals in each generation either by cloning non-Unc hermaphrodites when the arrays permitted XX viability or crossing non-Unc males into *him-5 unc-76; xol-1* hermaphrodites and recovering non-Unc XO progeny bearing the arrays, if the arrays caused XX lethality.

Once array lines were made with *xol-1* transgene deletion derivatives in the *him-5 unc-76; xol-1* strain, they were crossed into *yIs44(fox-1); him-5 unc-76; xol-1* hermaphrodites. *yIs44* has multiple copies of the *fox-1*-containing cosmid R04B3 and the *rol-6* plasmid pRF4, which causes the array animals to have a roller phenotype. *xol-1* activity was scored by the presence of non-Unc males, an indication of rescue of the XO-specific lethality caused by the *xol-1(y9)* mutation. DNA sequences and details of transgene constructions are available upon request for pMN66 (Δ introns II-VI), pMN67 (Δ intron VI), pMN60 (Δ introns II-V), and pCP5 (intron VI in 3' UTR).

The *lacZ* reporters *Pxol-1::lacZ::xol-1 3'UTR* (pMN27) and *Pxol-1::lacZ::intron VI::unc-54 3' UTR* (pMN110) were co-injected at 20 µg/ml with 150 µg/ml of *unc-76*(+) marker plasmid p76-16b into *him-5(e1490) unc-76(e911)* hermaphrodites. Array lines were crossed into *yIs44(fox-1); him-5 unc-76* hermaphrodites. pMN27 was made from the promoterless *lacZ* vector pRD95.11 from A. Fire. pMN110 was made by inserting intron VI into pMN21, a *Pxol-1::lacZ* transcriptional fusion made from the intron-rich *lacZ* plasmid pPD95.03 from A. Fire. DNA sequences and details of reporter transgene constructions are available upon request. Embryonic β-galactosidase expression was assayed as previously described using 4 µl of 4% 5-bromo-4-chloro-3-indolyl-D-galactopyranoside staining solution and incubation overnight at room temperature (*Nicoll et al., 1997*). More than 1000 embryos of each genotype were scored for β-galactosidase expression.

## Isolation of total RNA from embryos

Total RNA was isolated from 0.2 to 0.5 ml of washed, packed embryos that had been stored at −80°C. Frozen embryos were ground to a very fine powder using a liquid-nitrogen-cooled mortar and pestle, adding more liquid nitrogen as needed to prevent the material from thawing. Ground, frozen material was vortexed in two microfuge tubes, each with 1 ml Trizol, for 30 s to 1 min and then incubated for 5 min at room temperature. Chloroform (0.2 ml) was added to each tube, vortexed for 30 s, and incubated at room temperature for 2–3 min. After centrifugation at 13,000 × g for 5 min at 4°C, the upper aqueous phases (0.75 ml/tube) were transferred to a new tube, and 0.5 ml of the material was distributed into each of three tubes. An equal volume of phenol was added to each tube, vortexed, and 0.2 ml $CHCl_3$ was added and vortexed. Following centrifugation at 13,000 × g for 5 min at 4°C, the aqueous phases were transferred to new tubes, and an equal volume of $CHCl_3$ was added and vortexed. Following a 13,000 × g spin in a microfuge for 5 min at 4°C, the aqueous phases were transferred to new tubes and incubated a room temperature with one volume of 2-propanol for 10 min. Following centrifugation at 13,000 × g spin for 10 min at 4°C, the supernatants were removed, and the pellets were washed with 70% ethanol and dried in a Spin-Vac. Each pellet from the three tubes was resuspended in 100 µl water, and all volumes were combined into one tube (~0.3 ml total). One volume of 4 M LiCl was added and incubated overnight at 4°C. After a 13,000 × g microfuge spin for 5 min at 4°C, the aqueous phase was transferred to a new tube. The RNA was incubated with 0.1× volume of 3 M NaOAc and 2.5× volume of ethanol on ice for more than 10 min. After centrifugation at 13,000 × g for 10 min at 4°C, the aqueous phase was removed, the pellet was washed with 70% ethanol, and dried in a Spin-Vac. The RNA was resuspended in 100 µl water, and the O.D. was read after diluting a sample 250-fold.

## Quantification of transcript levels

Quantitative RT-PCR was used to measure transcript levels in RNA isolated from three independent growths of the four strains listed in *Figure 2—source data 1* as previously described (*Gladden and Meyer, 2007*; *Van Gilst et al., 2005*). Three different sets of RT-PCR experiments were performed with each of the three independent growths of worms. Worms were grown on egg plates (http://www.wormbook.org) prior to isolating the mixed-stage embryos, and the total RNA was isolated as described above. A complete list of primer sequences is in *Supplementary file 1*. The *xol-1* primers were designed to measure all splice variants simultaneously, not just the active ones.

Transcript levels for *xol-1* and *nhr-64* were normalized to the transcript level of *fasn-1*, which is expressed constitutively throughout embryogenesis, by adjusting the Ct (cycle threshold) value of *fasn-1* measured in each strain to equal the Ct value of *fasn-1* measured in the control embryos, either *him-5* or wild type. This adjustment equalizes the small variations in concentration of the starting material added to each PCR reaction from different RNA preparations. The transcript level of each mutant strain was then expressed as fold change relative to the control embryos, either *him-5* or wild-type (ΔCt). The normalized Ct value for each transcript measured in each strain was subtracted from the normalized Ct value of the same transcript measured in control embryos. The difference between these values corresponds to the change in transcript levels relative to those in control animals. Ct values are expressed as PCR cycle numbers. Each PCR cycle increases the concentration of the template by twofold. Therefore, to convert the difference in Ct values to a relative change in concentration, the expression $2^{\Delta Ct}$ was used. The same protocol was used to measure *xol-1* and *fasn-1* transcript levels in the four strains of *Figure 2—source data 1* when they were normalized to the *nhr-64* transcript.

## Riboprobe preparation for RNase protection assays and FOX-1 cross-linking experiments

Riboprobes were made from linearized pBluescript SK plasmids in which DNA encoding the RNA region of interest had been cloned adjacent to the T3 promoter. For a 20 µl T3 RNA polymerase reaction mixture, the following reagents were combined: 4 µl of 5× transcription buffer (Stratagene), 2 µl each of 50 mM ATP, GTP, and CTP, 10 µl of α-$^{32}$P-UTP (Amersham, 400 Ci/mmol, 10 µCi/µl), 1 µl linearized DNA (0.4 µg), 1 µl of 0.2 M DTT, 1 µl RNasin (Promega 40 U/µl), and 1 µl T3 RNA Polymerase (Stratagene, 50 U/µl). The reaction was incubated for 60 min at 40°C. The following reagents were added to the transcription reaction and incubated for 15 min at 37°C: 1 µl RNasin, 2.5 µl vanadyl ribonucleoside complex (200 mM), 20 µl water, 6 µl 5× transcription buffer, and 1 µl RNase free DNase I (1 mg/ml). The reaction was extracted with 50 µl phenol. The phenol was back extracted with 50 µl TE. The two aqueous phases were combined and extracted with 100 µl of chloroform:isoamyl alcohol (24:1). Two ethanol precipitation steps were then performed. For the first, the following reagents were added and the mixture was incubated at room temperature for 5 min and then spun: 1 µl glycogen, 66 µl of 5 NH$_4$Ac, and 450 µl of 100% ethanol. The pellet was resuspended in 100 µl water. For the second, 10 µl of 3 M NaOAc and 250 µl of 100% ethanol were added, and the mixture was spun and dried. The pellet was resuspended in 100 µl water.

## RNase protection assays

RNase protection assays were performed using Ambion's RPA (RNase Protection Assay) III Kit.

## PCR assay showing that only intron VI is retained in strains with high FOX-1 levels

To determine how many introns were retained in *xol-1* when FOX-1 was at high levels, we performed PCR on cDNA prepared from total RNA isolated as for the RPA. Using primers overlapping the first ATG of *xol-1* (5'-gcaggttgaagcaaattctgagagaag-3') and within the 7th exon (5'-cactcttcatcctcatcatacgtg tc-3'), only two bands corresponding to the size of the 2.5 kb and 2.2 kb transcripts were amplified from wild-type, *him-5*, and *yIs44; him-5* animals. The primer set does not detect cDNA from the alternatively spliced 1.5 kb transcript.

## DNA sequence analysis of clones from *xol-1* or *lacZ* transcripts

To characterize splice variants of *xol-1* transcripts made in low and high levels of FOX-1, we synthesized cDNA and then cloned and sequenced *xol-1* specific transcripts. To capture full-length transcripts, the first strand of cDNA was made from total embryonic RNA made from *him-5* and *yIs44 (fox-1); him-5* strains using a Clontech Advantage RT-for-PCR Kit and an oligo(dT)$_{18}$ primer. We then performed two sets of PCR reactions to capture transcripts that included exon 6 of *xol-1* and the downstream splice variants. In the first PCR reaction, the *xol-1*-specific primer in exon 6 (CSNP-8, 5'-GAGTTTGATAGCCAAGTTGCTCTTG-3') was used in combination with a 3' RACE primer with a unique tag on it (5'-aagcagtggtatcaacgcagagTAC(T)$_{30}$N-1 N-3', where N-1 is A,C, or G and N is A, C, G, or T). In the second PCR reaction, a nested exon 6 primer closer to the 3' end of the exon (CSNP-9, 5'-CATGAGCAAGTAGAAGGTTTCGAAG-3') was used in combination with a primer to the unique tag (5'-aagcagtggtatcaacgcagagT-3'). The PCR products were TOPO cloned (Thermo Fisher Scientific) and sequenced.

To characterize the effect of different FOX-1 levels on splice variants from the *Pxol-1-* driven *lacZ* reporter carrying intron VI of *xol-1,* we made total embryonic RNA from two strains: TY2882, *him-5 (e1490) unc-76(3911) V; yEx280* and TY3082, *yIs44; him-5(e1490) unc-76(3911) V; yEx280. yEx80* is an extra-chromosomal array that carries the *unc-76* co-injection marker and the reporter *Pxol-1::lacZ::xol-1 intron VI::unc-54 3' UTR* (pMN110). The first strand of cDNA was made using a Clontech Advantage RT-for-PCR Kit and the 3' RACE primer with the unique tag from above (5'-aagcagtggtatcaacgcagagTAC(T)$_{30}$N-1 N-3'). In the first PCR reaction the *lacZ*-specific primer just before intron VI (CSNP-12, 5'-AATCAGGCCACGGCGCTAATCACG-3') was used in conjunction with a primer to the unique tag (5'-aagcagtggtatcaacgcagagT-3'). In the second PCR reaction, a nested *lacZ*-specific primer closer to intron VI (CSNP-13, 5'-CGCTGGATCAAATCTGTCGATCC-3') was used in conjunction with primer to the unique tag. PCR products were TOPO cloned and sequenced. In the second PCR, because the sequence from the beginning of intron VI to the poly A tail was too long to pick up transcripts with intron VI properly spliced out in strain TY2882, a third set of PCR reactions was performed using exclusively *lacZ* sequences that flank intron VI. DNA from the first PCR reaction was used to perform PCR with the nested lacZ-specific primers CSNP-14 (5'-CTAATCACGACGCGCTGTATCG-3') and CSNP-15 (5'-GTCGGCAAAGACCAGACCGTTC-3'). PCR products were TOPO cloned and sequenced. The expected PCR product with no intron VI splicing was 900 bp, and the expected product with correct intron VI splicing was 434 bp.

## Cross-linking protocol

A cocktail of 2 μl normalized riboprobe, 2 μl of water or RNA competitor, and 1 μl non-competing cold background RNA (intron III, 100 molar excess, 217 ng) was added to 1.5 ml tubes on ice. Riboprobe was used in RNA excess, and intron VI riboprobe was determined empirically to be in excess when $8 \times 10^6$ cpm riboprobe was cross-linked with 32 ng FOX-1 protein. Riboprobes of intron VI subregions (A–E) or intron VI with small deletions were cross-linked at the same molar concentration as full-length intron VI (determined by percent of labeled Us in each fragment vs. Us in intron VI). A cocktail of the following reagents was added to the chilled tubes: 11 μl of cross-linking buffer [20 mM HEPES (pH 7.9), 20% glycerol, 100 mM KCl, 0.2 mM EDTA, 0.2 mM PMSF, 0.5 mM DTT, 0.091 ng/μl BSA], 2 μl of 20 mg/ml heparin, and 2 μl of 16 ng/μl FOX-1 protein (or a series of FOX-1 concentrations). The 20 μl reactions were incubated for 25 min in a room-temperature water bath. The tubes were placed in an aluminum block pre-chilled on ice and then UV irradiated for 12 min (model UVP-54G shortwave UV lamp: 254 nm, 1 cm from tube). Reactions were digested for 15 min at room temperature with RNase (1 μl of 3.5 mg/ml RNase A, final concentration of 175 ng/μl and 1 μl of 10 U/μl RNase T1, final concentration of 0.5 U/μl). Then 7 μl of SDS sample buffer and 1 μl of β-mercaptoethanol (0.5 M final) were added to the samples. The samples were boiled for 5 min, spun down in a microfuge, and half of the sample (15 μl) was loaded onto a NOVEX 10% Tris-glycine gel, which was run at 150 V for 90 min. The gel was dried for at least 30 min and exposed to a phosphor screen. The $^{32}$P RNA-protein bands were imaged using a Fuji Phosphorimager and quantified using the MacBASV2.5 software package.

For RNA competition experiments using $^{32}$P-labeled intron VI as the riboprobe, the probe included all 466 nt of intron VI and was transcribed from pMN147. The *xol-1* intron III non-competing cold background RNA included all 456 nt of intron III and was transcribed from pKA1. The RNA

oligos B-37, C-35, E-35, and E-25 were purchased from Dharmacon Research, Inc The sizes of the cold RNA competing fragments for each region in *Figure 5A* and their fraction of Us relative to intron VI are the following: A, 124 nt (0.27); B, 104 nt (0.17); C, 116 nt (0.23); D, 89 nt (0.16); E, 137 nt (0.21); intron VI with ΔB-37 and ΔC-35, 426 nt (0.87); intron VI with ΔB-37, ΔC-35, and ΔE26, 400 nt (0.83); and intron III, 456 nt (0.88).

FOX-1 protein was tagged at the N-terminus with maltose binding protein (MBP), expressed in bacteria, and purified with an amylose resin column, using 10 mM maltose to elute FOX-1 from the resin.

## Genome editing using CRISPR/Cas9

The endogenous *fox-1* and *xol-1* genes (*Figure 6*) were edited using microinjected Cas9 ribonucleoprotein complexes (RNPs), as described previously (*Farboud et al., 2019*). Target-specific CRISPR guide RNAs (crRNA) (*Supplementary file 2*) and trans-activating crRNA (tracrRNA) were obtained from Dharmacon. crRNAs have a 2xMS modification to improve nuclease resistance. For all experiments, a co-CRISPR strategy was used that relied on the *dpy-10* co-conversion marker (*Supplementary files 2–4*) to enrich for edited worms, based on a Dpy or Rol phenotype, prior to PCR screening and DNA sequencing to identify desired mutations (*Arribere et al., 2014*; *Kim et al., 2014a*; *Ward, 2015*).

To edit *fox-1*, two crRNAs (*Supplementary file 2*) targeted Cas9-dependent DSBs to two sites flanking *fox-1*. A single-stranded DNA oligonucleotide (ssDNA) (*Supplementary file 3*) with homology 5' of one DSB and 3' of the other served as the homology-directed repair template to join the broken DNA ends and thereby delete *fox-1*. The ssDNA repair template was injected with Cas9 RNPs at a final concentration of 0.5 μM. PCR using the three primers BF-2492, BF-2493, and BF-2494 in a single reaction (*Supplementary file 4*) permitted the simultaneous detection of an unedited wild-type *fox-1* gene, a heterozygous 10 kb *fox-1* deletion, and a homozygous 10 kb *fox-1* deletion. PCR amplification with BF-2493, which anneals to sequences targeted for deletion, and BF-2494, which anneals to sequences downstream of the deletion, yields a 307 bp amplicon only from an unedited *fox-1* gene. In contrast, PCR amplification with BF-2492 and BF-2494, which anneals to sequences upstream of the deletion, yields a 475 bp amplicon only from a deleted *fox-1* gene, named *fox-1(y793)*. Precise excision was confirmed by PCR amplifying the edited locus with primers BF-2492 and BF-2494 and performing Sanger sequencing with primer BF-2492.

To edit *xol-1* DNA encoding intron VI, two crRNAs targeted Cas9-dependent DSBs to sites flanking intron VI (*Supplementary file 2*). The broken DNA ends were joined by homology-directed repair that was templated from exogenously provided double-stranded DNA (dsDNA) fragments, either 1553 bp for the deletion allele *xol-1(y810)* or 1651 bp for all others. The dsDNA repair templates with intron VI mutations used in *Figure 6* were synthesized as gBlocks by Integrated DNA Technologies. The templates also included silent mutations to eliminate the PAM (AGG to AGA at codon 330 associated with crispr_bf69 RNA) or to mutate a key nucleotide (GCA to GCT at codon 255 associated with crispr_bf68 RNA). These mutations prevented Cas9 from cleaving the repair template and from re-cleaving the precisely edited genomic locus. They did not alter the primary amino acid sequence of *xol-1*. The dsDNA repair template also included approximately 500 bp of uninterrupted homology on either side of the silent mutations. Sequences of dsDNA repair templates are available upon request. The gBlocks were subcloned into pCR-Blunt II vectors, sequenced, and then PCR amplified using Phusion High-Fidelity DNA polymerase with primers BF-2507 and BF-2508 (*Supplementary file 4*). The resulting amplicon was purified and concentrated using Qiagen MinElute columns. dsDNA repair templates were injected with Cas9 RNPs at a final concentration of 350 ng/μl. Resulting co-converted Dpy or Rol F1 animals were lysed, and the *xol-1* intron VI region examined. To screen for all *xol-1* intron VI mutations except the *y810* deletion, primers BF-2518 and BF-2519 were used to PCR amplify the targeted locus, and BF-2518 was used to perform Sanger sequencing to confirm precise editing. To screen for the *xol-1(y810)* deletion, F1 Dpy or Rol worms were screened by PCR using primers BF-2301, BF-2676, and BF-2746 in a single reaction to detect an unedited gene and heterozygous or homozygous deletion variants lacking the intron. Strains carrying homozygous deletions were reexamined by amplifying *xol-1* using BF-2518 and BF-2519, and performing Sanger sequencing with primer BF-2301 to confirm precise editing. All oligonucleotides used to screen for CRISPR/Cas9 induced mutations and to determine sequences of resulting mutations are listed in *Supplementary file 4*.

Sequences within intron VI of *xol-1* include five potential FOX-1 binding sites (*Figure 6*), each having GCAUG and/or GCACG motifs. The motifs are distributed throughout the intron and start at the following positions relative to the 5′ end of the intron: 124 nt, 152 nt, 231 nt, 257 nt, and 440 nt.

Adjacent and overlapping the GCACG motif at position 152 is a GCUUG sequence resembling a FOX-1 binding motif. The overlap is one nt: GCUUGCACG. In vitro binding studies of Rbfox to GCAUG and GCACG sites showed that GCACG is a higher affinity binding site than GCUUG. Since this site can only be occupied by a single FOX-1 protein, and FOX-1 is more likely to bind GCACG than GCUUG, we classify this entire GCUUGCACG sequence as a GCACG site (*Figure 6* and *Figure 6—figure supplement 1*). Even though the GCUUG motif is highly unlikely to bind FOX-1, we mutated it in several strains in case it can bind FOX-1 when GCACG is altered. To disrupt it in *xol-1 (y803)* and *xol-1(y804)*, the GCUUG motif was mutated to AUUUA in addition to changing the GCACG motif, making the final sequence AUUUAUACA. For *xol-1(y808)*, the GCUUG motif was mutated to GCUUA in addition to mutating the GCACG motif, making the final sequence GCUUAUACA. For *xol-1(y809)*, the GCUUG motif was mutated to AUUUG while preserving the adjacent GCACG motif, making the final sequence AUUUGCACG. For *xol-1(y807)*, the GCUUG site was mutated to GCUCG, while preserving the GCACG motif, making the final sequence GCUCG-CACG. The GCUUG motif was not mutated by itself to measure the effect of GCUUG loss because changing that motif to AUUUA would mutate the adjacent higher affinity GCACG motif.

## Quantifying the effects on viability and *xol-1* splicing repression caused by mutations in endogenous FOX-1 regulatory sequences induced using CRISPR/Cas9

### Viability of *xol-1* mutant XX animals

The viability of wild-type and *xol-1* mutant XX animals in *Figure 6* was determined by the following protocol: individual adult hermaphrodites were placed onto NG agar plates with thin OP50 bacterial lawns and transferred to new plates three times per day for 4 days. Laid embryos were counted after hermaphrodites were transferred. During the 2–5 days after embryos were laid, viable adult progeny were counted. Percent viability was calculated by the formula: (number of adults/number of embryos) $\times$ 100. Counts were compiled from at least two independent experiments. Statistical comparisons of viability used the two-tailed unpaired Student's t-test. The same procedure was used for determining the viability of *fox-1(y793)* XX hermaphrodites.

### Viability of mutant XX animals in combination with *sex-1(RNAi)*

For experiments in *Figure 6*, *Figure 6—figure supplement 1*, and *Figure 6—source data 1*, RNAi against the XSE gene *sex-1* was performed as described previously (*Gladden et al., 2007*; *Kamath et al., 2001*) using RNAi clones from the Ahringer RNAi feeding library (*Kamath and Ahringer, 2003*). L1 stage wild-type and mutant XX animals were grown on plates with a lawn of *E. coli* that produced *sex-1* dsRNA until they reached adulthood. Individual adult hermaphrodites were then transferred to new RNAi plates with a lawn of dsRNA-producing *E. coli* three times a day for 4 days. Laid embryos were counted after hermaphrodites were transferred. During the 2–5 days after embryos were laid, viable adult progeny were counted. Percent viability was calculated by the formula: (number of adults/number of embryos) $\times$ 100. Counts were compiled from at least three independent experiments for *Figure 6* and *Figure 6—figure supplement 1* and from two for *Figure 6—source data 1*. Statistical comparisons of viability used the two-tailed unpaired Student's t-test.

### Viability of *xol-1* mutant XO animals and *fox-1* mutant XO animals

To determine the viability of *xol-1* mutant XO males, mutant XX hermaphrodite strains of genotype *him-5(e1490)* V; *xol-1* X were constructed. *him-5(e1490)* XX animals produce 33% XO male progeny (*Hodgkin et al., 1979*). Individual *him-5(e1490)*; *xol-1* young adult XX hermaphrodites were placed on NG agar plates with a thin OP50 lawn and transferred to new plates three times per day for 4 days. Laid embryos were counted after hermaphrodite transfer. During the 2–5 days after embryos were laid, plates were examined, and viable adult XX and XO progeny were counted. Percent viability for *him-5(e1490)* XO and for *him-5(e1490)*; *xol-1* XO animals was calculated by the formula: [(number of adult males/number of embryos) / (0.33)] $\times$ 100.

The same protocol was used to determine the viability of *fox-1(y793 null)* XO males, except the starting strain was *him-8(e1489); fox-1(y793)*. Because our control *him-8(e1489)* hermaphrodites produce 36% XO male progeny, the percent viability for *him-8(e1489); fox-1(y793)* XO males was calculated by the formula: [(number of adult males/number of embryos) / (0.36)] × 100.

## Viability of *xol-1* mutant XO animals with elevated FOX-1 levels

To determine the viability of *xol-1* XO mutant animals with elevated FOX-1 levels, mutant XX hermaphrodite strains of genotype *yIs44 IV; him-5(e1490) V; xol-1 X* were constructed. *yIs44* is an integrated transgenic array carrying multiple copies of the *fox-1*-containing cosmid R04B3 and the *rol-6* plasmid, pRF4, which causes animals to have a Rol phenotype. Individual *yIs44; him-5(e1490); xol-1* young adult hermaphrodites were placed on NG agar plates with thin OP50 lawns and transferred to new plates three times per day for 4 days. After the hermaphrodites were transferred, laid embryos were counted. During the 2–7 days after embryos were laid, plates were examined, and viable adult XX and XO progeny were counted. Percent viability of *yIs44; him-5(e1490)* XO animals and *yIs44; him-5(e1490); xol-1* XO mutants was calculated by the formula: [(number of adult males/number of embryos) / (0.33)] × 100.

## *asd-1* extra-chromosomal transgenic arrays

To overexpress *asd-1(+)* in wild-type nematode strains, extra-chromosomal transgenic arrays carrying multiple copies of *asd-1(+)* were made by coinjecting a PCR product containing the endogenous *Pasd-1::asd-1::asd-1* 3′ UTR sequences (50 ng/μl) and plasmid pRF4 [dominant *rol-6(su1006)*] (100 ng/μl) into *him-8(tm611)* hermaphrodites. Individual Rol worms were picked to isolate and maintain three independent array-bearing lines. Primers used to amplify the genomic *asd-1* locus (BF-2748: 5′-agatttgatcattttgtgcaggaactccttcgttatttgcctggactac-3′ and BF-2749: 5′-gatgtgcagctattttgagatttccgatgcctgatttagatgatgagccgatggatg-3′) amplified sequences 1936 bp upstream of the *asd-1* translation start site through 613 bp downstream of the translation terminator, adjacent to the proceeding ORF.

## Acknowledgements

We thank D Black, T Blumenthal, D Rio, and members of the Meyer lab for critical discussions, T Cline for critical comments on the manuscript, and D Stalford for assistance with figures. Some strains were provided by the CGC, which is funded by the NIH Office of Research Infrastructure (P40 OD010440).

## Additional information

### Funding

| Funder | Grant reference number | Author |
| --- | --- | --- |
| Howard Hughes Medical Institute | Senior Investigator Award | Barbara J Meyer |
| National Institutes of Health | R 35 GM 131845 | Barbara J Meyer |

The funders had no role in study design, data collection and interpretation, or the decision to submit the work for publication.

### Author contributions

Behnom Farboud, Conceptualization, Data curation, Formal analysis, Investigation, Methodology, Writing - review and editing; Catherine S Novak, Data curation, Investigation, Methodology; Monique Nicoll, Conceptualization, Data curation, Investigation, Methodology; Alyssa Quiogue, Investigation; Barbara J Meyer, Conceptualization, Resources, Formal analysis, Supervision, Funding acquisition, Writing - original draft, Project administration, Writing - review and editing

Author ORCIDs
Behnom Farboud https://orcid.org/0000-0001-7439-3095
Barbara J Meyer https://orcid.org/0000-0002-6530-4588

Decision letter and Author response
Decision letter https://doi.org/10.7554/eLife.62963.sa1
Author response https://doi.org/10.7554/eLife.62963.sa2

## Additional files

### Supplementary files

• Supplementary file 1. List of primers. This table includes a complete list of primers.

• Supplementary file 2. List of target-specific sequences for guide RNAs used in CriSPR/Cas9 genome editing experiments. This table lists the gene targets, the target-specific sequence for each Cas9 guide RNA, the figure in which the results are presented, the genomic coordinates corresponding to each guide RNA, and the reference name of each guide. Two guides were used together to delete the *fox-1* gene, and two guides were used together to replace sequences within intron VI of *xol-1*. Coordinates are based on Wbcel235/c11 version of the *C. elegans* genome.

• Supplementary file 3. DNA sequences of repair templates used for CRISPR/Cas9 genome editing experiments. This table presents information for CRISPR/Cas9 genome editing experiments involving HDR using single-stranded repair templates. The table lists the gene targets that were edited, the figure in which the results are presented, the DNA sequences of the repair template, with SNPs highlighted in red letters and deletion junctions highlighted in blue letters, the reference names of related guides, and the reference names of the oligonucleotide repair template. Two guide RNAs were used together in combination with the repair template to delete the endogenous *fox-1* gene, resulting in the allele *fox-1(y793)*. The double-stranded repair templates used for inducing mutations in the DNA encoding intron VI of endogenous *xol-1* are available upon request but are too long to include in this table.

• Supplementary file 4. List of oligonucleotides used to screen for CRISPR/Cas9 induced mutations and to determine sequences of resulting mutations. The table lists the gene targets, the figure or table in which the results are presented, the sequences of oligonucleotides used to screen for CRISPR/Cas9 induced mutations or to determine sequences of resulting mutations, the reference name of each oligonucleotide, and the function of each oligonucleotide. For *asd-1*, the oligos were used to verify the construction of strains built with a pre-existing *asd-1* mutation, not for identifying new mutations made using Cas9. The oligonucleotides BF-2507 and BF-2508 were used to synthesize *xol-1* repair templates rather than to identify Cas9-induced mutations in intron VI.

• Transparent reporting form

### Data availability

All data generated or analysed during this study are included in the manuscript and supporting files.

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
