## [Decision Letter]

**Acceptance summary:**

This study demonstrates how the gene dose of the RNA binding protein *Fox-1* determines expression of the sex determining gene *Xol-1* in developing nematodes. The authors show how the variable concentration of *Fox-1* arising from one or two X chromosomes, combined with repetitive binding elements of varying affinity within the *Xol-1* transcript, allow the splicing of *Xol-1* RNA to sensitively respond to the sex genotype. Previous studies have shown how transcription factors can act to count X chromosomes and determine sex. This work shows that RNA binding proteins can also report chromosome dose.

**Decision letter after peer review:**

Thank you for submitting your article "Action of an RNA Binding Protein to Amplify the Dose-Sensitive X-Chromosome Signal that Determines *C. elegans* Sex" for consideration by *eLife*. Your article has been reviewed by two peer reviewers, and the evaluation has been overseen by Douglas Black as Reviewing Editor and James Manley as the Senior Editor. The following individual involved in review of your submission has agreed to reveal their identity: Christopher B Burge (Reviewer #2).

The reviewers have discussed the reviews with one another and the Reviewing Editor has drafted this decision to help you prepare a revised submission.

Summary:

In this study, Farboud and colleagues explore the contribution of the RNA binding protein FOX-1 to sex determination in *C. elegans*, via alternative splicing of transcripts from the *xol-1* gene, a master sex determination switch. Using a series of intron-deletion strains, the authors determined that FOX-1 regulates *xol-1* levels in sex determination via FOX-1-regulated alternative splicing of *xol-1* intron VI, and characterize the splice isoforms generated by FOX-1 binding at different levels of the RBP via RT-PCR. Five (GCAUG and GCACG) FOX-1 motifs are identified as functional target motifs using in vitro binding assays. Finally, a series of CRISPR/Cas9 strains confirm their findings in vivo and explores the spectrum of alternative splicing regulation exerted by different levels of FOX-1 and its target motifs in *xol-1* intron VI.

The reviewers agreed that the experiments are well-designed and thorough, and support the proposed model and described mechanism of FOX-1 regulation of *xol-1* splicing in this important developmental process. The dose-dependent regulation of splicing by FOX-1 also has implications for cell- and tissue-specific splicing regulation. Several changes and additions to the study are needed to clarify and support the author's claims regarding sufficiency of regulation and the importance of FOX-1 dose in their mechanism.

Essential revisions:

1) While both mechanisms the authors suggest are likely at play, it seems more important to keep FOX-1 at single copy in males than it is to have two copies of FOX-1 in hermaphrodites. This would be consistent with the fact that loss of *fox-1* does not seem to impact hermaphrodites, but having a duplication of *fox-1* in XO animals leads to lethality and feminization, as has been described in earlier papers characterizing the genetics of fox-1. The FOX-1 dependent inactivation of residual *xol-1* activity in hermaphrodites may be more dispensable than the authors present. The discussion seems to spend more time discussing the role of FOX-1 in repressing *xol-1* activity in hermaphrodites rather than its single copy requirement in males. The significance of FOX-1 dose in males should be equally presented.

2) The data presented in Figures 2-4 require better quantification and presentation of replicates (Figure 2: % viability as in Figure 6 rather than "VIA/DEAD," Figure 3: all gels visible and replicates quantified, and Figure 4: quantified and/or replicates shown in the supplement). In addition, the changes in isoform ratio described in Figure 3 by RNase protection assays need to be assessed in relation to the absolute levels of the transcripts compared to a control gene.

3) *xol-1* expression is regulated both transcriptionally (by XSEs and ASEs) and post-transcriptionally (by at least FOX-1). What are the respective contributions to *xol-1* regulation by transcriptional and post-transcriptional regulation in wild-type worms? How does this change among some of the genetic contexts described? To some extent, the former could be estimated simply by qRT-PCR of ASE/XSE KOs and the latter could be assessed by changes in expression of ASEs and XSEs upon FOX-1 depletion or overexpression.

4) The characterization of the sensitivity to FOX-1 levels and motif abundance in *xol-1* intron VI in Figure 6 is fairly compelling, but additional synthetic genetic experiments could greatly strengthen this relationship. For example, can the authors show that there is improved viability with low-affinity (GCUUG) motifs in a high FOX-1 background?

5) At the end of the Results and in the Discussion, the authors make a distinction between their dosage-sensitive mode of FOX-1 regulation and that of mammalian Rbfox, which sometimes only requires a single motif to regulate splicing. However, it has been known for many years that multiple GCAUG motifs yield a stronger enhancer in mammalian cells (Modafferi, 1999), and more recent studies have shown that mammalian Rbfox regulation is sensitive to both protein concentration and motif binding affinity (Begg et al., 2020). The distinction between the two systems is thus not that clear. If the authors wish to clearly demonstrate a functional difference between worm and mammalian Rbfox proteins, perhaps a rescue experiment replacing *fox-1* with a mammalian homolog would be informative.

---

## [Author Response]

Essential revisions:1) While both mechanisms the authors suggest are likely at play, it seems more important to keep FOX-1 at single copy in males than it is to have two copies of FOX-1 in hermaphrodites. This would be consistent with the fact that loss of fox-1 does not seem to impact hermaphrodites, but having a duplication of fox-1 in XO animals leads to lethality and feminization, as has been described in earlier papers characterizing the genetics of fox-1. The FOX-1 dependent inactivation of residual xol-1 activity in hermaphrodites may be more dispensable than the authors present. The discussion seems to spend more time discussing the role of FOX-1 in repressing xol-1 activity in hermaphrodites rather than its single copy requirement in males. The significance of FOX-1 dose in males should be equally presented.

"Viability of the *sex-1(RNAi)* XX animals declined more than 10-fold, from 44% to 3% (p = 0.002), when the dose of *fox-1* was reduced by half, from two copies to one copy, demonstrating dose-dependence of FOX-1 function in XX animals (new Figure 7A,B). Similarly, viability of *sex-1(RNAi)* XX animals declined from 44% to 7% (p = 0.006) when all FOX-1 binding sites in intron VI were mutated in one copy of *xol-1* (new Figure 7A,C). As expected, viability of *sex-1(RNAi)* XX animals declined to an intermediate level when either the three GCACG motifs (18%) (p = 0.018) or the two GCAUG motifs (13%) (p = 0.012) were mutated in one copy of *xol-1* (new Figure 7A,D,E). Viability of *sex-1(RNAi)* XX animals with five low-affinity GCUUG motifs in one copy of *xol-1* was better (36%) than that with either two GCAUG (p = 0.023) or three GCACG (p = 0.017) mutated motifs in one *xol-1* copy (new Figure 7F). Hence, the dose-dependence and necessity of FOX-1 function in regulating alternative *xol-1* splicing in XX animals is evident both from reducing the dose of the *trans*-acting FOX-1 protein or by mutating different combinations of *cis*-acting FOX-1 binding sites in only one copy of *xol-1*. Thus, FOX-1 acts as an X-signal element to convey X-chromosome number by regulating alternative *xol-1* splicing in a dose-dependent manner.” Having two doses of *fox-1* in XX embryos is as important for viability as restricting the dose of *fox-1* to one in XO embryos. See last paragraph of the Results section.

Our new experiments are fully consistent with prior genetics experiments demonstrating the cumulative, dose-dependence action of X-signal elements, including FOX-1, in both XX and XO animals. For example, deleting one copy of *ceh-39* and *sex-1* from XX animals caused no lethality, but deleting one copy of *ceh-39*, *sex-1*, and *fox-1* killed more than 70% of XX animals, and deleting one copy of all genetically identified XSEs killed all XX animals (Akerib and Meyer, 1994; Carmi and Meyer, 1999; Farboud et al., 2013; Gladden et al., 2007). In reciprocal experiments, duplicating one copy of *fox-1* killed 25% of XO animals, while duplicating *fox-1* and *ceh-39* killed 50% of XO animals, and duplicating one copy of all genetically identified XSEs killed all XO animals (Akerib and Meyer, 1994; Carmi and Meyer, 1999; Nicoll et al., 1997). These original genetic observations are now described in the Introduction of our revised manuscript to set the stage properly for our paper.

We also performed additional genome editing experiments to evaluate the performance of low-affinity GCUUG binding motifs vs. high affinity GCAUG and GCACG in regulating *xol-1* in XX and XO animals with different levels of FOX-1 (Figure 6 and the new Figure 6—figure supplement 2). The results described below address point 4 requested by the reviewers.

2) The data presented in Figures 2-4 require better quantification and presentation of replicates (Figure 2: % viability as in Figure 6 rather than "VIA/DEAD," Figure 3: all gels visible and replicates quantified, and Figure 4: quantified and/or replicates shown in the supplement). In addition, the changes in isoform ratio described in Figure 3 by RNase protection assays need to be assessed in relation to the absolute levels of the transcripts compared to a control gene.

For Figure 2, the extra-chromosomal array assay of wild-type and mutant *xol-1* transgenes to define regions of *xol-1* necessary for FOX-1 repression is efficient and highly reliable. It is intended as a qualitative assay to define regions of importance rather than as a quantitative assay as in Figure 6, which can assess viability with the expectation of normal Mendelian inheritance for mutant chromosomes. Extra-chromosomal arrays can be lost through mitotic divisions and are not expected to show Mendelian inheritance, hence the use of visible phenotypic markers to track whether and where an animal has the array or not. All conclusions derived from these rapid array assays were verified and pursued in great detail in the paper through multiple lines of experiments to define the mechanism of FOX-1 action.

For this array assay, a cocktail of wild-type or mutant *xol-1* transgenic DNA is co-injected with wild-type unc-76 marker DNA into a *him-5* strain that also carries knock-out mutations in *xol-1* and *unc-76*. The starting strain produces viable Unc XX animals and dead XO animals. Progeny XX animals that have received a *xol-1*(+) *unc-76(+)* array will be non-Unc, and progeny XO animals will be viable and non-Unc. The arrays are established in a *him-5(*null) *unc-76*(null); *xol-1*(null) mutant strain that has normal levels of FOX-1 and then crossed into a *yIs44(fox-1); him-5*(null) *unc-76* (null); *xol-1*(null) mutant strain that produces high FOX-1 levels.

Because XX animals are highly sensitive to the dose of *xol-1*, we had to use a concentration of wild-type *xol-1* DNA (5 µg/ml) that was 30-fold lower than the concentration typical for most markers, including the *unc-76*(+) marker injected at 150 µg/ml. Of the 7 independent extra-chromosomal arrays made with *xol-1*(+) DNA, all had to be maintained in XX strains with two wild-type copies of *fox-1*, otherwise the XX animals would die, and the lines could only be maintained through male progeny of genetic crosses. This XX-specific property and the fact that multiple copies of *fox-1* on a stable integrated chromosomal transgenic array [(*yIs44fox-1)*] could fully repress *xol-1*(+) transgenic arrays and cause complete XO-specific lethality made this design a perfect qualitative assay to determine *xol-1* regions that define the sites of FOX-1 action.

While *xol-1*(+) arrays allowed XX and XO animals to be viable, arrays of *xol-1* transgenes that either lacked all introns (∆ intron II-VI, 3 independent lines) or just intron VI (∆ intron VI, 3 independent arrays) caused XX animals to die (> 99%) or to have severe dosage compensation defects (< 1%), indicating that critical FOX-1 regulatory sites were missing. Only loss of intron VI was required to kill XX animals, demonstrating its importance.

In contrast, the XO animals with these arrays were viable, and the lines could only be maintained through array-bearing XO males in genetic crosses. For each line, resulting crosses yielded hundreds non-Unc males, indicating the transgene had *xol-1*(+) activity, but NO or extremely rare and sterile dosage-compensation-defective XX animals. When all 6 arrays were crossed into strains with high FOX-1 levels, hundreds of wild-type males were produced for each line but No or only rare sterile hermaphrodites, further indicating that FOX-1 regulatory sequences had been removed. The same result held for 2 arrays of *xol-1* transgenes that had intron VI relocated from its normal site in the coding gene into the 3' UTR of *xol-1* without splice junctions.

Critically, 5 independent arrays of *xol-1* transgenes that kept intron VI but lacked introns II-V (∆ intron II-V) behaved like *xol-1*(+) arrays, indicating that introns II-V are not critical for FOX-1 repression. With normal FOX-1 levels hundreds of XX and XO animals were viable from each line maintained through hermaphrodites. With high FOX-1 levels, hundreds of XX animals were viable from each line but NO males survived.

This qualitative assay showed that intron VI is the critical intron to mediate FOX-1 repression of *xol-1*. Further quantification would not change the conclusion about the central importance of intron VI or add any new useful information about FOX-1-mediated repression. Subsequent experiments in the paper, including (i) RNase protection assays and cDNA sequencing to demonstrate the *xol-1* isoforms and relative prevalence in normal or high FOX-1 levels; (ii) transgenic studies showing the sufficiency of intron VI; (iii) extensive FOX-1 binding studies that define FOX-1 binding motifs exclusively in intron VI but no other introns; (iv) genome editing studies to demonstrate the importance and dose-sensitivity of FOX-1 motifs in intron VI, all serve to confirm and extend the conclusions in Figure 2.

Response for Figure 3: For the RNase protection assays presented in the original submission (Figure 3 and Figure 3—figure supplement 1), and for the additional gels shown in the revised manuscript (Figure 3—figure supplement 1D), the RNA samples for comparison [*him-5* XX and XO embryo populations] vs. [*yIs44(fox-1);him-5* XX and XO embryo populations] were first quantified and tested with the *act-1* (actin) gene as a control. An *act-1* RPA gel for these two classes of samples is now shown in Figure 3—figure supplement 1D (left). The quantified RNA samples were then tested with *xol-1* probes as listed in Figure 3 and Figure 3—figure supplement 1 to assess the relative levels of the 2.2 kb (active), 2.5 kb (inactive), and 1.5 kb (inactive) *xol-1* isoforms in the presence of normal or high levels of FOX-1.

The ratio of 2.2 kb and 2.5 kb splice isoforms or the ratios of 2.2 kb, 2.5 kb, and 1.5 kb splice isoforms were then calculated from isoforms within a single RNA population. Ratios of isoforms from different populations were then compared to assess the change in isoform ratios caused by varying the FOX-1 level. Because the quantification of *xol-1* isoforms for each calculated ratio is from within the same RNA (and same RPA reaction), it seems that direct comparison of *xol-1* isoforms is an appropriate calculation without normalizing each isoform to the level of a control *act-1* gene. We compare ratios of *xol-1* isoforms across RNA populations, not levels of individual *xol-1* isoforms across different RNA samples. The *act-1* controls allow us to know that the quantity and quality of RNAs used in RPA reactions is similar.

In Figure 3—figure supplement 1D (right) we now show an RPA gel quantifying the 3 *xol-1* variants using a different probe from that used in Figure 3A. Further quantification or repetition of RPAs will not change or main conclusion that FOX-1 promotes the formation of the 1.5 kb and 2.5 kb isoforms but not the 2.2 kb isoform. As a further, higher resolution examination of *xol-1* isoforms than the RPAs, we show in the original and current manuscript, the DNA sequence analyses of the cloned *xol-1* cDNA isoforms in normal and high FOX-1 levels (Figure 3B,C). The cDNA sequences reinforce and extend the conclusions from the RPAs.

Response for Figure 4: For Figure 4A,B, quantification of the analysis showing necessity and sufficiency of intron VI for FOX-1-mediated *xol-1* repression is presented in the revised text and in the revised figure legend. We assayed β-galactosidase activity in multiple lines of two different *lacZ* reporters: (1) five independent extra-chromosomal array lines of a control reporter in which the *xol-1* promoter directed *lacZ* expression and the 3' UTR was from *xol-1* and (2) four independent extra-chromosomal lines of a reporter in which *xol-1*'s intron VI was placed into an exon of a *lacZ* reporter gene driven by the *xol-1* promoter with a 3' UTR from the control myosin gene *unc-54*.

All nine independent lines were established in XX animals and then crossed into a *him-5* strain that produces XX and XO embryos with normal levels of FOX-1 and into a *him-5* strain that produces XX and XO embryos with high levels of FOX-1 from *yIs44(fox-1).* For each of these nine lines, we assayed at least 1000 progeny embryos for each of the different genotypes (wild-type XX, *him-5* XX and XO, as well as *yIs44(fox-1); him-5* XX and XO). In total, we assayed over 27,000 embryos. The images in Figure 4A,B are highly representative of the embryos for each genotype derived for each of the two different classes of reporters (with or without *xol-1* intron VI).

As a further, higher resolution examination of the splicing patterns for the intron VI-containing *lacZ* reporter in animals with normal or high FOX-1 levels, we show cDNA sequence analysis of cloned *lacZ* transcripts from embryos of each genotype (*him-5* or *yIs44(fox-1);him-5)* (Figure 4C,D). The cDNA sequences reinforce and extend the conclusions from the β-galactosidase assays: High FOX-1 levels cause non-productive splicing of intron VI and hence loss of β-galactosidase coding capacity.

3) xol-1 expression is regulated both transcriptionally (by XSEs and ASEs) and post-transcriptionally (by at least FOX-1). What are the respective contributions to xol-1 regulation by transcriptional and post-transcriptional regulation in wild-type worms? How does this change among some of the genetic contexts described? To some extent, the former could be estimated simply by qRT-PCR of ASE/XSE KOs and the latter could be assessed by changes in expression of ASEs and XSEs upon FOX-1 depletion or overexpression.

Transcriptional regulation is the primary form of *xol-1* repression. Mutations that knock out both copies of the *sex-1* nuclear hormone receptor XSE gene cause 80% of XX hermaphrodites to die. Combination of the *sex-1* knock out and the knock out of both copies of *fox-1* causes 100% of hermaphrodites to die, demonstrating the importance and cumulative effect of both transcriptional and splicing regulation in repressing *xol-1* in XX animals. As shown in our manuscript, the XX-specific lethality caused by reducing *sex-1* activity can be suppressed by multiple copies of *fox-1*, demonstrating that loss of transcriptional repression can be compensated for by enhancing the repression achieved by alternative splicing via FOX-1.

We have rewritten the discussion to address point 5 requested by the reviewers.

4) The characterization of the sensitivity to FOX-1 levels and motif abundance in xol-1 intron VI in Figure 6 is fairly compelling, but additional synthetic genetic experiments could greatly strengthen this relationship. For example, can the authors show that there is improved viability with low-affinity (GCUUG) motifs in a high FOX-1 background?

In our original submission, we demonstrated through genome editing experiments using combinations of motif mutations and different doses of *fox-1* (both normal one or two doses or high doses) that multiple high-affinity GCAUG and GCACU motifs are essential for FOX-1 regulated alternative *xol-1* splicing in XX and XO animals (Figure 6).

In our new experiments, we showed that multiple low-affinity GCUUG motifs in intron VI are not sufficient to repress *xol-1* with the normal FOX-1 levels present in wild-type XX or XO animals but are sufficient for *xol-1* repression in both sexes with high FOX-1 levels. We present additional experiments in Figure 6 for XO animals and an additional set of experiments for XX animals presented in the new Figure 6—figure supplement 2.

As described above, we have now included better quantification, explanation, and presentation of replicates for data in Figures 2-4 to address point 2 requested by the reviewers.

5) At the end of the Results and in the Discussion, the authors make a distinction between their dosage-sensitive mode of FOX-1 regulation and that of mammalian Rbfox, which sometimes only requires a single motif to regulate splicing. However, it has been known for many years that multiple GCAUG motifs yield a stronger enhancer in mammalian cells (Modafferi, 1999), and more recent studies have shown that mammalian Rbfox regulation is sensitive to both protein concentration and motif binding affinity (Begg et al., 2020). The distinction between the two systems is thus not that clear. If the authors wish to clearly demonstrate a functional difference between worm and mammalian Rbfox proteins, perhaps a rescue experiment replacing fox-1 with a mammalian homolog would be informative.

Our Discussion now includes the following section:

“Mechanisms underlying the action of FOX family members in regulating alternative pre-mRNA splicing

Multiple high-affinity FOX-1 binding motifs are necessary to restrict the non-productive mode *xol-1* splicing to XX embryos over the small FOX-1 concentration range that distinguishes XX from XO embryos. […] Five GCUUG motifs are sufficient to permit high FOX-1 levels to suppress the XX-specific lethality caused by *sex-1(RNAi)* or to kill all XO embryos. These results converge with and extend those found for mammalian Rbfox.”